# Miocene Volcanism in the Slovenský Raj Mountains: Magmatic, Space, and Time Relationships in the Western Carpathians

Rastislav Demko [1], Marián Putiš [2,*], Qiu-Li Li [3], David Chew [4], Lukáš Ackerman [5] and Ondrej Nemec [2]

[1] State Geological Institute of Dionýz Štúr, Mlynská Dolina 1, SK-817 04 Bratislava, Slovakia; rastislav.demko@geology.sk

[2] Department of Mineralogy, Petrology and Economic Geology, Faculty of Natural Sciences, Comenius University, SK-842 15 Bratislava, Slovakia; ondrej.nemec@uniba.sk

[3] State Key Laboratory of Lithospheric Evolution, Institute of Geology and Geophysics, Chinese Academy of Sciences, Beijing 100029, China; liqiuli@mail.iggcas.ac.cn

[4] Department of Geology, School of Natural Sciences, Trinity College Dublin, D02PN40 Dublin 2, Ireland; chewd@tcd.ie

[5] Institute of Geology of the Czech Academy of Sciences, CZ-165 00 Prague, Czech Republic; ackerman@gli.cas.cz

* Correspondence: marian.putis@uniba.sk

**Abstract:** The Miocene volcanic-intrusive complex in the Slovenský Raj Mountains, middle Slovakia, comprises a swarm of subalkaline basalts and basaltic andesites with alkaline basalts, trachybasalts and basaltic trachyandesites. Basaltic to doleritic feeder dykes and sporadic hyaloclastite lavas are exposed in contact with the Triassic Bódvaszilas Formation of the Silica Nappe. The primary clinopyroxene, plagioclase, and Fe-Ti oxide assemblage also contains calcite spheroids inferred to represent carbonatitic melt. These spheroids are associated with subsolidus chlorite, actinolite, magnetite, titanite, calcite, and epidote. Micropoikilitic clinopyroxene, albite, and Ti-magnetite formed due to rapid quenching. There was an incorporation of host rock carbonate during the eruption. The erupted products are the result of magmatic differentiation of the parental basaltic tholeiitic magma with a redox of $\Delta QFM = +1$ to $+3$, affected by varying degrees of 0%–50% fractionation and the assimilation of carbonate material in a shallow magmatic reservoir. REE geochemistry shows N-MORB-like type patterns with both $La_N/Yb_N$ and $La_N/Sm_N < 1$ at near constant $Eu/Eu^*$ (~0.9). This is supported by $\varepsilon Nd_{(t=13\ Ma)}$ values of +8.0 to +7.4 determined from the basaltic rocks. The REE values can be modeled by 1% fractional melting of garnet peridotite mixed with 7% melting of spinel peridotite of PM composition (1:9 proportions). SIMS and LA-ICP-MS U/Pb analysis of zircons yields a concordant age of $12.69 \pm 0.24$ Ma and a $13.3 \pm 0.16$ Ma intercept (Serravallian) age. The Middle Miocene volcanic activity was related to subduction-collision processes along the boundary of the Cenozoic ALCAPA (Alps–Carpathians–Pannonia) microplate and the southern margin of the European plate.

**Keywords:** Miocene; N-MORB-like basalts; continental transtension; Western Carpathians; geochemistry; melt simulations; U-Pb zircon geochronology





## 1. Introduction and Geological Setting

Basaltic rocks can represent indicators of different geodynamic settings within the continental or oceanic crust, e.g., [1–7]. The chemical composition of basalts is controlled by the composition of the mantle source, the extent of melting, and subduction-related metasomatic processes, which all affect the LILE–HFSE–REE systematics, enabling discrimination of a particular geodynamic regime [6,8,9].

Our work documents hitherto unknown Miocene basaltic volcanism in the Inner Western Carpathians (IWC), which accompanied regional coeval calc-alkaline volcanism. This complex regional volcanic system, likely with differing mantle sources, requires at

least a partial modification of existing geodynamic models [10,11]. We present petrological and geochemical information on igneous rocks from a Miocene volcano-intrusive complex that formed along the Muráň fault system in the Slovenský Raj Mountains (SRM) Lesnica Valley, middle Slovakia (Figure 1). The basalt geochemistry shows an affinity to MORB ocean floor basalts ($La/Yb_c$ and $La/Sm_c < 1$) but a closer affinity to basalts that formed in transform fault environments (e.g., $Sm/Yb_c > 1$). These characteristics likely imply these LREE-depleted basalts are indicators of a continental tectonic setting without active spreading. The basalt magma assimilated carbonate material from the host limestones during eruptions. The assimilated carbonate affected the solidification and led to the crystallization of magmatic albite and the formation of inferred carbonatitic melt. We also compare our geochronological data with the temporal evolution of magmatic activity in the Western Carpathians segment of the Cenozoic ALCAPA (Alps–Carpathians–Pannonia) microplate and the adjacent European platform/plate (EP), which shows a migration of volcanic activity from west to east.

### 1.1. Regional Cenozoic Tectonics and Magmatism

The Cenozoic volcanism of the Western Carpathians (WC) is linked to the formation of the Carpathian arc and a system of continental back-arc basins resulting from the collision of Africa with the EP. The Eocene continent–continent collision of ALCAPA and the Tissia microplates caused lateral movement of ALCAPA following the closure of the Pieniny Magura Ocean. South-directed subduction of oceanic or mafic continental lithosphere under the northwestern, northern, and finally, northeastern ALCAPA margin led to the formation of an accretionary prism of oceanic flysch sediments in the frontal part of the arc. The translation of ALCAPA during the collision with the EP margin was controlled by slab roll-back, retreat, and break-off (delamination) of the subducted crust, e.g., [10,11]. The collision resulted in the thrusting of the accretionary prism sediments over the EP and the formation of the Silesian and Magura napes of the Outer Western Carpathians (OWC) Flysch Zone. Contemporaneous formation of the Pieniny Klippen Belt (PKB) resulted in tectonically detached Mesozoic carbonate blocks strung out along the OWC and Inner Western Carpathians (IWC) boundary. The continental collision was accompanied by a 30°–50° anticlockwise rotation of the ALCAPA block and the opening of the Pannonian intra-arc sedimentary basins [10–13].

Neogene to Quaternary volcanism between ~21 and 0.1 Ma [14] of the WC is concentrated in the Central Slovakia Volcanic Field (CSVF) and chains of stratovolcanoes in the Slanské vrchy and Vihorlat mountains (Figure 1). The Neogene calc-alkaline volcanism is effusive and rich in pyroclastics. Andesite magma fractionated to dacite and rhyolite, which is typical of volcanic provinces at convergent lithospheric plate boundaries. However, in the WC region, they are linked to extensional tectonics accompanied by the formation of the intracontinental back arc Pannonian basins [10–20]. Calc-alkaline andesite-rhyolite volcanism was followed by alkaline basaltic volcanism from Late Miocene to Quaternary times at 8–0.2 Ma [14,16,17,21].

The petrogenesis of the andesites and their parental mafic magmas in the CSVF results from the interaction between magmas derived from an enriched mantle source that was previously metasomatized by fluids from subducted sediments and subsequent interaction and mixing with melts derived from the lower continental crust. A significant change occurred before 13 Ma, when calc-alkaline magmas were derived from a more enriched, FOZO-like source, indicating a heterogeneous upper mantle source composition [19,20]. Similarly, the petrogenesis of the post-collisional calc-alkaline magmas of Vihorlat and the Ukrainian Carpathians volcanic chains shows that the magma composition is influenced by variable enrichment of a heterogeneous mantle source with subduction-derived fluids and subsequent contamination by upper crustal material [22].

The basic 8–0.2 Ma sodium-alkaline magmatism (Figure 1), which followed the calc-alkaline magmatism, has been variously regarded as a product of low degree partial melting of the upper asthenosphere during the post-extensional stage [19,20], the diapiric

rise of the mantle in back-arc basins [12], small-scale mantle plumes emplaced during post-collision [16], or by the extraction of mantle magmas from a metasomatized asthenospheric source through deep-seated faults in a compressional tectonic regime [23].

The Neogene volcanic fields include small volcanic provinces in the OWC north of the PKB, which are represented by intrusions in the Magura flysch sediments of the obducted accretionary prism (Figure 1). These include trachyandesites [24], biotite-amphibole-pyroxene (Bt-Amp-Px) andesites [25], and trachybasalts to trachyandesites [26,27] in Moravia Uherský Brod in the Czech Republic (No. 9 in Figure 1). Other intrusive bodies in the Polish sector of the PKB are found near the Wzar and Bryjarka mountains and Jarmuta, where Miocene basanites [28], andesites [29], and calc-alkaline basaltic andesites and andesites [30] were intruded into sandstones of the Magura flysch zone (No. 10 in Figure 1) along with intrusions of Amp-Px andesites into Paleogene sandstones of the Magura flysch in Ladomirov in the eastern Vihorlat Mountains [25] (No. 11 in Figure 1). The petrogenesis of these magmas is related to the melting of variably metasomatized subcontinental lithospheric mantle [27,30] impinging on the base of the lithosphere with emplacement potentially controlled by trans-lithospheric NW–SE-trending fault zones, as suggested by [26].

Our field and petrographic investigations focused on the discovery of volcanic rocks within sandstones and carbonates of the Triassic Bódvaszilas Formation (asterisk in Figure 1A,B) in the Silica Nappe [31]. This formation contains alkaline and tholeiite dolerites and lavas, which represent feeder dyke systems as small relics of effusive hyalo-clastite basalts. A Triassic to Jurassic age for the feeder dyke complex was initially proposed based on the similar structural position of Triassic volcanic rocks of the Meliata Unit, e.g., ref. [32] and references therein in the hanging wall of the Silica Nappe. This work provides the first detailed descriptions of the volcanic intrusive complex from the Lesnica Valley and includes field, petrographic, geochemical, and geochronological investigations. We describe the volcanic and magmatic solidification history and petrogenesis of this complex and consider its wider spatial and temporal relations with the volcanism of the WC arc.

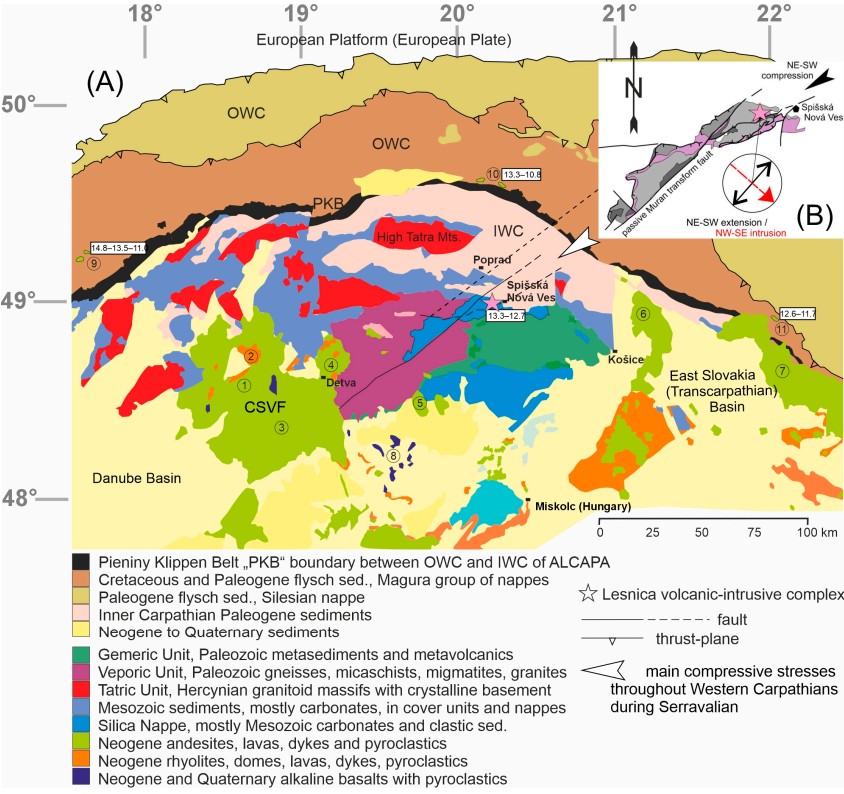

**Figure 1.** (**A**) Geological map of the Western Carpathians arc and adjacent areas in ALCAPA with the distribution of Neogene volcanic mountains highlighted (compiled from [33,34]). Ages of the small

volcanic provinces in the Outer Western Carpathians (OWC) are from [18] and from this study in the Silica Nappe in the Slovenský Raj Mountains (SRM) in the Inner Western Carpathians (IWC) and are marked by boxes. (**B**) Tectonic structures affecting the Triassic carbonate blocks (grey) and sandstones—claystones of the Bódvaszilas Formation (purple), showing NE–SW compression and local NE–SW extension in the SRM during the Serravallian [10,15] which controlled magmatic activity in the Lesnica Valley. Key: ① Štiavnica stratovolcano, ② rhyolites of Jastrabská Formation, ③ Javorie stratovolcano and volcanoes of the Krupina plain, ④ Poľana stratovolcano, ⑤ Vepor stratovolcano, ⑥ Slanské vrchy, ⑦ Vihorlat mtn., ⑧ alkaline basalts and basanites, and volcanic-intrusive complexes from OWC: ⑨ Moravia, Uherský Brod area in Czech Republic, ⑩ Pieniny Klippen Belt, ⑪ Ladomir complex. CSVF—Central Slovakia Volcanic Field.

### 1.2. Spatial and Temporal Tectonomagmatic Evolution of the Western Carpathians

Cenozoic volcanism (16.5–8.5 Ma) in the Carpathian–Pannonian region (CPR) was related to the subduction of the (North Penninic equivalent) Magura Ocean crust, the collision of ALCAPA with the EP, and the development of continental back-arc sedimentary basins [12]. Subaerial volcanism created several large volcanic edifices in the CPR, including the Kremnica Mountains, Štiavnica, Poľana, and Javorie stratovolcanoes, and the Slanské and Vihorlat mountains. Syntheses on the timing of Cenozoic volcanism are provided in [14,18,35–39].

Significant volcanism occurred in the Central Slovakia Volcanic Field (CSVF). The emplacement of a subvolcanic diorite intrusive complex in the Štiavnica stratovolcano is dated at 13.5–12.9 Ma [35] and was followed by the formation of calc-alkaline andesites. The stratovolcano erupted clinopyroxene-orthopyroxene-amphibole (Cpx-Opx-Amp) andesites at 13.7–13.6 Ma and terminated with explosive caldera formation and eruptions of rhyodacites at 13.2–12.8 Ma [40,41]. The voluminous production of calc-alkaline andesites in the Javorie stratovolcano was followed by a decrease in volcanic activity at 13.6 Ma and by erosion of the caldera [42]. These events indicate a decrease in activity or destruction of the stratovolcanic bodies in the CSVF at ca. 13 Ma.

There was a tectonic control on lithospheric mantle melting during collision of the WC with the EP, with a migration in magmatic activity from west to east: in Uherský Brod, Moravia, two phases of magmatism at 15–13 and 13.5–11.0 Ma [24,25], in the PKB at 13.3–10.8 Ma [29,43,44], and in Ladomirov in eastern Slovakia at 12.4 Ma [18,25,36–39].

The progressive west–east interaction between the colliding ALCAPA and EP blocks is documented by structural analysis of the Magura flysch sediments of the OWC [10,45], analog modeling [46], and analysis of the tectonic evolution of the WC [10,13,15] and the northern part of the Pannonian sedimentary deposits [11,13].

The migration of volcanic activity in the Western and Eastern Carpathians seems to be closely related to these tectonic processes [10,12,18,44,47,48]. Based on the first occurrence of volcanism in Moravia (13.5 Ma), the PKB (13.3 Ma), and Ladomirov (12.4 Ma) in the OWC, the effect of the collision between the WC and EP blocks on lithospheric melting can be observed. Magmatic events at 13.3–12.7 in the Lesnica Valley are synchronous with andesite intrusions in the PKB to the NNW, and there is likely a common tectonic origin for these two magmatic events. The collision of the IWC into the EP at 13.3 Ma coincides with the exhumation of the adjacent Vysoké Tatry (High Tatra) Mountains ([49] and references therein) between 20 and 7 Ma with a change from contraction in the Early Miocene to strike-slip movements in the Middle to Late Miocene. Paleotectonic reconstructions of the ALCAPA block [10,11,13] show a NE direction of ALCAPA and the IWC (17–15 Ma), which switches north in the Middle Miocene (15–13 Ma), with a NE–SW maximum compression orientation and subsequent NW–SE extension at the Middle/Late Miocene boundary (12–10 Ma). The west–east migration of compressive stresses along the collision boundary between the EP and the ALCAPA blocks is synchronous with thrusting in the Magura nappe and the termination of sediment deposition in the foreland Molasse Zone [10].

## 2. Materials and Methods

The samples are dolerite dykes (Les-1, Les-1A, RD2–17, RD5-07: N 48°55.126′ E 20°27.097′; Les-2: N 48°55.142′ E 20°27.144′), basaltic lavas (Les-3: N 48°54.964′ E 20°26.884′; Les-4: N 48°55.192′ E 20°27.239′) and basaltic hyaloclastite (Les-5: N 48°55.082′ E 20°26.992′) from the Lesnica Valley in the SRM in middle Slovakia. Fresh samples were collected, and any partly weathered rims were cut off to expose fresh internal portions for geochemical analyses.

Mineral composition and textures of rocks were investigated in polished sections by optical light microscopy. The chemical compositions of minerals were acquired using wavelength-dispersive (WDS) analyses on a CAMECA SX-100 electron probe microanalyzer (EPMA) in the State Geological Institute of Dionýz Štúr in Bratislava, Slovakia. Analytical conditions were a 15 keV accelerating voltage and a 20 nA beam current. The beam diameters varied between 1 μm and 5 μm, depending on the size of the analyzed target and the mineral type. Measured lines, crystals, and spectrometers were: Na (Kα TAP Sp1), Si (Kα TAP Sp1), Al (Kα TAP Sp1), Mg (Kα TAP Sp1), K (Kα LPET Sp3), Ca (Kα LPET Sp3), Fe (Kα LLIF Sp4), Sr (Lα LPET Sp3), F (Kα LPC0 Sp2), Cl (Kα LPET Sp3), Mn (Kα LLIF Sp4), Cr (Kα LLIF Sp4), Ni (Kα LLIF Sp4), Ti (Kα LPET Sp3), Zn (Kα LLIF Sp4), V (Kα LLIF Sp4) and for standards Na (albite), Si and Ca (wollastonite), Al ($Al_2O_3$), Mg (forsterite), K (orthoclase), Fe (fayalite), Sr ($SrTiO_3$), F ($CaF_2$), Cl (NaCl), Mn (rodonite), Cr (Cr), Ni (Ni), Ti ($TiO_2$), Zn (willemite), V (V).

Whole-rock powders were analyzed by X-ray fluorescence (XRF) for major elements and trace and rare earth elements (REE) in BUREAU VERITAS Ltd. laboratories in Vancouver, Canada. Inductively coupled plasma optical emission spectrometry (ICP-OES) and inductively coupled plasma mass spectrometry (ICP–MS) were used for whole-rock (including REE) geochemical analyses.

Zircon crystals were extracted using standard density and magnetic separation techniques. The zircon crystals were mounted in epoxy, polished, and coated with high-purity gold to reach <20 Ω resistance prior to analysis. High-quality images of zircon crystals under translucent and reflective light and SEM-CL (scanning electron microscope cathodoluminiscence) were acquired to guide spot choice and to avoid fractures and inclusions.

Measurements of U, Th, and Pb isotope compositions of zircons were performed on a Cameca IMS-1280HR SIMS at the Institute of Geology and Geophysics, Chinese Academy of Sciences in Beijing. The complete instrument description and analytical procedure for zircon dating are described in [50]. The primary beam of $O^{2-}$ was accelerated at 13 kV and focused to an ellipsoidal spot of approximately $20 \times 30$ μm in size. Positive secondary ions were extracted with a 10 kV potential. A mass resolution of approximately 5400 defined at 10% peak height separated Pb+ peaks from isobaric interference. A single electron multiplier in ion-counting mode measured secondary ion beam intensities by peak jumping mode. Pb/U calibration was performed relative to the Plešovice zircon reference material ($^{206}Pb/^{238}U$ age = 337.1 Ma, [51]), and U and Th concentrations were calibrated against zircon reference material 91500 (Th = 29 ppm, and U = 81 ppm, [52]). The Qinghu zircon reference material was analyzed as an unknown to monitor external uncertainties. Twenty-two Qinghu zircon measurements yielded a Concordia age of 160 ± 1 Ma, which is within error of the recommended value of 159.5 ± 0.2 Ma [53]). Concordia age diagrams were generated by the Isoplot/Ex v. 2.49 program [54].

LA-ICP-MS U-Pb age zircon data were acquired using a Photon Machines Analyte Excite 193 nm ArF excimer laser-ablation system with a HelEx 2-volume ablation cell coupled to an Agilent 7900 ICP-MS at the Department of Geology, Trinity College, Dublin. The instrument was tuned using NIST612 standard glass to yield Th/U ratios of unity and low oxide production rates (ThO+/Th+ typically <0.15%). A quantity of 0.4 l min$^{-1}$ He carrier gas was fed into the laser cell, and the aerosol was subsequently mixed with 0.6 l min$^{-1}$ Ar make-up gas and 11 mL min$^{-1}$ N2. Data reduction of the raw U-Pb isotope data was performed through the "VizualAge" data reduction scheme [55] in the IOLITE package [56]. Sample-standard bracketing was applied after the correction of downhole fractionation to account for long-term drift in isotopic or elemental ratios by normalizing

all ratios to those of the U-Pb reference standards. A repetition rate of 11 Hz and a circular spot of 24 μm were employed. Eleven isotopes ($^{49}$Ti, $^{91}$Zr, $^{175}$Lu, $^{202}$Hg, $^{204}$Pb, $^{206}$Pb, $^{207}$Pb, $^{208}$Pb, $^{232}$Th, $^{235}$U, and $^{238}$U) were acquired during each analysis, which comprised 27 s of ablation (300 shots) and 10 s of washout, the latter portions of which were used for the baseline measurement. 91500 Zrn ($^{206}$Pb/$^{238}$U TIMS age of 1065.4 ± 0.6 Ma [57] was used as the primary U-Pb calibration standard. The secondary standards GZ-7 zircon ($^{206}$Pb/$^{238}$U TIMS age of 530.26 Ma ± 0.05 Ma [58], Plešovice zircon ($^{206}$Pb/$^{238}$U TIMS age of 337.13 ± 0.37 Ma [51]) and WRS 1348 zircon ($^{206}$Pb/$^{238}$U TIMS age of 526.26 ± 0.70 [59]) yielded LA-ICP-MS ages of 533.1 ± 1.5 Ma, 335.9 ± 1.6 Ma, and 526.8 ± 2.4 Ma, respectively.

Strontium, Nd, and Pb isotopic compositions were acquired at the Institute of Geology of the Czech Academy of Sciences following the methods detailed in [60]. The protocol involves sample decomposition using a HF–HNO$_3$ mixture, separation of Sr, Nd, and Pb using ion exchange chromatography [61] and isotopic analyses ($^{87}$Sr/$^{86}$Sr, $^{143}$Nd/$^{144}$Nd, $^{206}$Pb/$^{204}$Pb, $^{207}$Pb/$^{204}$Pb, and $^{208}$Pb/$^{204}$Pb) using a Thermo Triton Plus thermal ionization mass spectrometer (TIMS, Thermo, Waltham, MA, USA). The external reproducibility of the Sr and Nd analyses was monitored by the measurements of the NIST SRM 987 and JNdi-1 solutions, respectively, with $^{87}$Sr/$^{86}$Sr of 0.710251 ± 0.000009 (2 s, $n$ = 6) and $^{143}$Nd/$^{144}$Nd of 0.512102 ± 0.000006 (2 s, $n$ = 6). Mass fractionation effects during lead isotopic measurements were corrected using an average mass fractionation factor of 0.136 per mass unit relative to the reference values [62].

Melt simulations were used to reconstruct magmatic fractionation and solidification under P-T-X conditions assumed during the solidification of basaltic melt. The rhyolite-MELTS package ver. 1.2.0. [63–65] was used to solve thermodynamic simulations with thermodynamic properties of the fluid phase from [66], allowing thermodynamic simulation of basalt solidification containing mixed H$_2$O-CO$_2$ fluid. A pressure of 1 MPa was employed. The temperature conditions of the simulations range between 1300 and 750 °C. REDOX conditions were simulated separately by fixation to ΔQFM = 0, ΔQFM = +1, ΔQFM = +2, and ΔQFM = +3.

Mineral abbreviations follow the IMA-CNMNC recommendations [67].

## 3. Results

### 3.1. Field Relationships and Petrography of the Basaltic Rocks

Basaltic blocks in limestone talus were discovered in the Lesnica River valley in the SRM during the fieldwork in [68]. Subsequent field work showed they represent a set of doleritic to basaltic dykes that intruded into sandstone and limestone. The significant relief of the Lesnica Valley with substantial limestone scree makes it difficult to follow the contacts of individual igneous bodies. Nevertheless, five ca. 1 m-wide intrusive dolerite bodies have been identified along the edge of the Lesnica River beneath the soil and talus. The bodies are emplaced within the Triassic sandstones of the Bódvaszilas Formation [31]. Basalts contain fine-grained fragments of decametric blocks of carbonate xenoliths that resemble local Jurassic limestone (Figure 2A) found within tectonic mélanges in the hanging wall of the Mesozoic Silica Nappe [33]. In addition to the shallow-intrusive dolerite bodies (Figure 2B), two basaltic bodies (Figure 2C) and basaltic hyaloclastite lavas (Figure 2D) were also identified. The close spatial association of effusive and subvolcanic bodies implies the shallow-intrusive dolerite bodies represent feeder channels for the basalt effusives, i.e., a subvolcanic to effusive complex of basalts.

The mafic subvolcanic rocks were classified as dolerites based on macroscopic (Figure 2B) and microscopic studies (Figures 3 and 4). Macroscopically, they are dark green massive rocks with visible (mm scale) pyroxene phenocrysts. Amygdaloids with chlorite or carbonate fill have been identified in the mineral matrix. Microscopic and BSE images of dolerites show subophitic (Figure 3A), graphitic (Figure 3B), and intersertal rock textures (Figure 3C) with some amygdales. The main rock-forming igneous minerals are Cpx, plagioclase (Pl), and Fe-Ti oxides. In the case of the dolerite sample, olivine (Ol) was identified based on pseudomorphs filled with chlorite (Chl)-serpentine minerals between unaltered Cpx and

saussuritized Pl. Most of the samples contain a Cpx-Pl-Ti-magnetite (TiMag) assemblage. The relationships between the main rock-forming minerals show different orders of crystallization between Cpx, Pl, and Fe-Ti oxides. Parts with porphyritic euhedral Cpx (Figure 3A) pass into portions with granophyric texture (Figure 3B), subophitic, and intersertal textures filled with a micropoikilitic Cpx-Pl-TiMag matrix (Figure 3C).

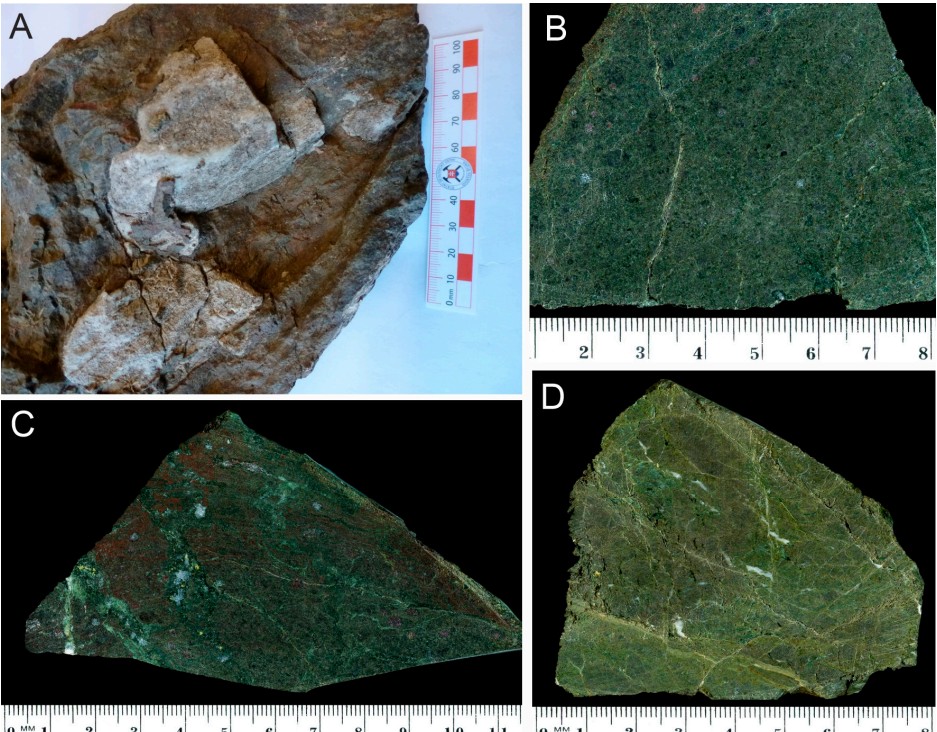

**Figure 2.** Field relationships of the basalts to the host rocks. (**A**) Pale carbonate fragments incorporated in basaltic dolerite during intrusion into the host limestone. This sample (Les-2) was used for LA-ICP-MS U-Pb zircon dating at Trinity College Dublin. (**B**) Les-1 dolerite sample used for SIMS dating in Beijing. (**C**) LES-3 basalt sample. (**D**) LES-5 basaltic hyaloclastite sample.

Petrographic relationships between mineral assemblages and their crystal habits indicate non-equilibrium crystallization, which was controlled by strong undercooling of the magma when emplaced in the host sediments. Cooling was enhanced by the small magma volume. The texture of the rock changes from holocrystalline subophitic to intersertal. The interstitial rock spaces are filled with the micropoikilitic Cpx-albite (Ab)-TiMag assemblage, which represents the solidification product of the residual interstitial melt.

The presence of basaltic lavas and hyaloclastites was identified via microscopy. The striking difference to the dolerites is the absence of Cpx phenocrysts, which are represented only in the form of microliths, often of anhedral habit. Albitic Pl chiefly defines acicular and thin-columnar crystals, often with skeletal habit (Figure 3D) caused by rapid crystallization during magma degassing and supercooling. The effusive basalt matrix contains small anhedral Cpx, Fe-Ti oxides, but especially minerals characteristic of subsolidus auto alteration (Chl-actinolite (Act)-Mag-titanite (Ttn)-epidote (Ep)-calcite (Cal). Compared to the dolerites, Pl crystallizes before Cpx.

The magmatic Cpx-Pl-TiMag association is in direct contact with the Chl-Act-Mag-Ttn ± Cal, Ep assemblage, which results from synvolcanic auto alteration due to the reaction of residual interstitial melt with $CO_2$-rich residual fluids under subsolidus conditions during advanced intrusion cooling and solidification. The Chl-Act-Mag-Ttn-Cal-Ep assemblage resembles that of greenschists or skarns that often develop between igneous rocks and carbonates (Figure 3E–H). Calcite is the dominant carbonate, forming thin rims on the

subsolidus mineral assemblage, often with Ep. Calcite and Ep together indicate the effects of iron oxidation by $CO_2$.

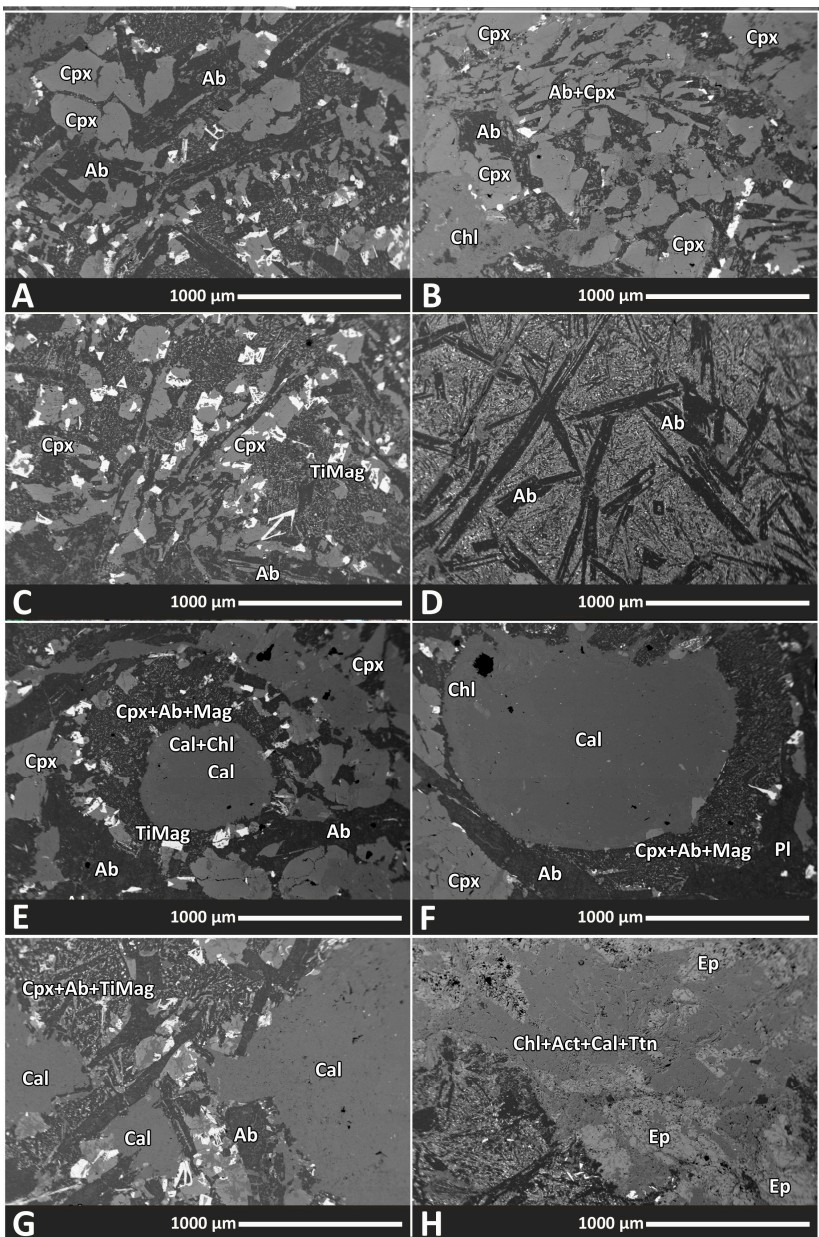

**Figure 3.** BSE images of basaltic textures. (**A**) Crystals of Cpx of augite (Aug)-diopside (Di) composition between lamellar albitic Pl and interstitial micropoikilitic Cpx-Ab-TiMag matrix. Skeletal Fe-Ti oxides from a transition horizon to the micropoikilitic matrix. (**B**) Ab-Cpx graphitic intergrowth demonstrating cotectic Ab-Cpx crystallization and a primary magmatic origin for the albitic Pl epitaxially oriented to the surrounding Cpx. The bright white crystals represent Fe-Ti oxides. (**C**) Cpx crystals in a graphitic intergrowth with Ab passing into acicular Ab. The amount of skeletal Fe-Ti oxides increases towards the periphery of the Ab-Cpx accumulation at the interface with the interstitial micropoikilitic Ab-Cpx-Mag matrix. (**D**) Albitic Pl crystals in a Cpx-Mag-Chl-Act-Cal matrix. The acicular and skeletal Ab indicates rapid crystallization. (**E**–**G**) Zoned spherical objects in the dolerite. The distribution of TiMag outside the main Ab and Cpx zone indicates later Fe-Ti oxide fractionation during the already crystallizing Ab-Cpx assemblage. (**H**) Poikilitic Ab-Cpx-TiMag association in the basalt enclosed in a subsolidus "skarn-like" assemblage of Chl-Act-Cal-Ttn-Ep.

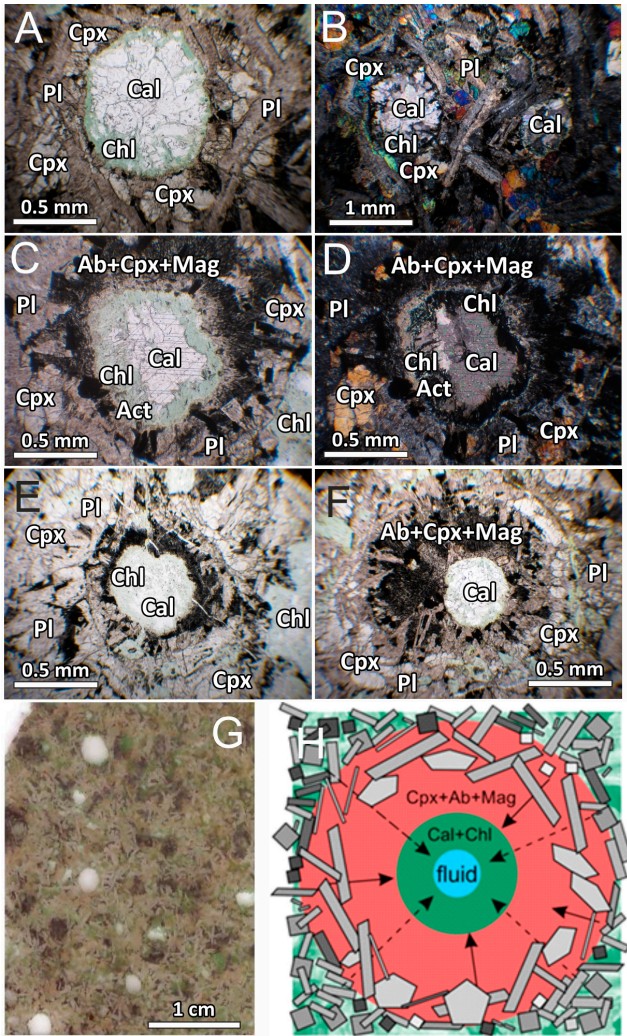

**Figure 4.** Polarized light microscopic images (**A**–**F**) document the evolution of spherical zoned aggregates "oceli". (**A**–**F**) Textural evidence of "oceli" representing late-magmatic to subsolidus process. Carbonatic (Cal-Chl?) "oceli" grew within a silicate Cpx-Pl-TiMag network (e.g., (**A**)). A fine-grained, rapidly quenched Cpx-Ab-TiMag micropoikilitic matrix (see also the BSE images in Figure 3) is found between the magmatic Cpx-Pl-TiMag silicate network and the "oceli" that are accompanied by subsolidus Chl-Act-Cal-Ep-Ttn aggregates. (**G**) Polished thin section of the Les-1 dolerite with a relatively homogeneous distribution of "oceli" varying in diameter from 2 to 5 mm. The pale Cal core is enclosed in dark spheroidal TiMag-bearing aggregates of Cpx and Ab. (**H**) Model reconstructing "oceli" formation based on fluid-driven melt extraction from the interstitial space of the solidifying basalt. The interstitial melts from the crystal mush infiltrate into the central part of the solidifying Cpx-Pl/Ab-Mag and Chl-Cal assemblages.

Spherical amygdale-shaped microscopic bodies (Figure 3E–G) with marked zoning are observed in dolerites in intersertal parts surrounded by crystal networks. Chlorite-Cal, or Cal-filled microspheric textures are observed. The more zonally complicated microspheric textures contain Cpx-Ab-TiMag micropoikilitic assemblages at the rims (Figure 3E–G), which are also identified separately in the micropoikilitic matrix. The micropoikilitic Cpx-Ab-Mag zones located on the outer edge of the spherical bodies pass into zones with the Cal core.

The spatial relationships between the micropoikilitic rim and the Chl-Cal core of the spherical "oceli" structures point to the physical interaction between silicate melt, possible carbonatitic melt, and a fluid phase. The Chl?-Cal in the central parts is the precipitation product of the inferred carbonatitic melt, which might have coexisted with the silicate melt

as represented by the micropoikilitic Cpx-Ab-Mag mass. The highly efficient extraction between Cpx-Ab-Mag and Chl?-Cal melts happens because of a strong viscosity contrast between these two coexisting melts.

The petrographical reconstruction of the "oceli" in Figure 4 is controlled by the degassing dynamics of the fluid phase and the interaction between the gas and the residual coexisting melts. The interaction between the gas and the interstitial melts only takes place in the zones with spherical objects, often described as "oceli". In parts without the development of spherical structures, the solidification of the interstitial melt proceeds to the crystallization of the micropoikilitic Cpx-Ab-Mag matrix or to the auto alteration "skarn" Chl-Act-Ttn-Ep-Cal assemblage (Figure 3H).

### 3.2. Mineral Chemistry and Oxygen Fugacity

The composition of igneous minerals reflects the composition of the magma at the time of intrusion and the evolution determined by the physical conditions. Since each of the samples represents a separate subvolcanic body or lava from a chemically differentiated common original magma, it is expected that these characteristics will be recorded by the evolution of the mineral compositions.

#### 3.2.1. Clinopyroxene

Clinopyroxenes (Table S1) follow the [69] classification and are mainly augite (Aug) and diopside (Di) with separate fractionation trends towards the wollastonite (Wo) and hedenbergite (Hed) fields (Figure 5). Augite is observed in subophitic dolerite pyroxenes RD-2/17 (Wo$_{43.1-38.2}$ enstatite En$_{49.9-48.3}$ ferrosilite Fs$_{12.0-6.9}$) or ophitic-graphitic pyroxenes from dolerite Les-1 (Wo$_{43.6-38.9}$En$_{53.9-43.6}$Fs$_{14.5-7.1}$). A different chemical evolution of Cpx can be observed in basaltic andesite Les-3 and some Cpx from dolerite Les-1A, where separate trends 1-, 2-, 3-, 4- diverge from the main differentiation path with decreasing Wo and increasing Fs content (Figure 5). A decrease in Wo is accompanied by an increase in Fe$^{3+}$ together with Na, as follows (1): Fe$^{3+}$ 0.06–0.1 and Na 0.02–0.05 *apfu*; (2): Fe$^{3+}$ 0.07–0.1 and Na 0.04 *apfu*; (3): Fe$^{3+}$ 0.06–0.1 and Na 0.03–0.05 *apfu*; (4): Fe$^{3+}$ 0.03–0.09 and Na 0.02–0.04 *apfu*. Some of the Les-1A pyroxenes reach Na > Fe$^{3+}$ *apfu* and trend towards Na-Ca pyroxene (aegirine Aeg$_{2.8-2.2}$ jadeite Jd$_{2.3-0.1}$Di$_{97.8-95.1}$).

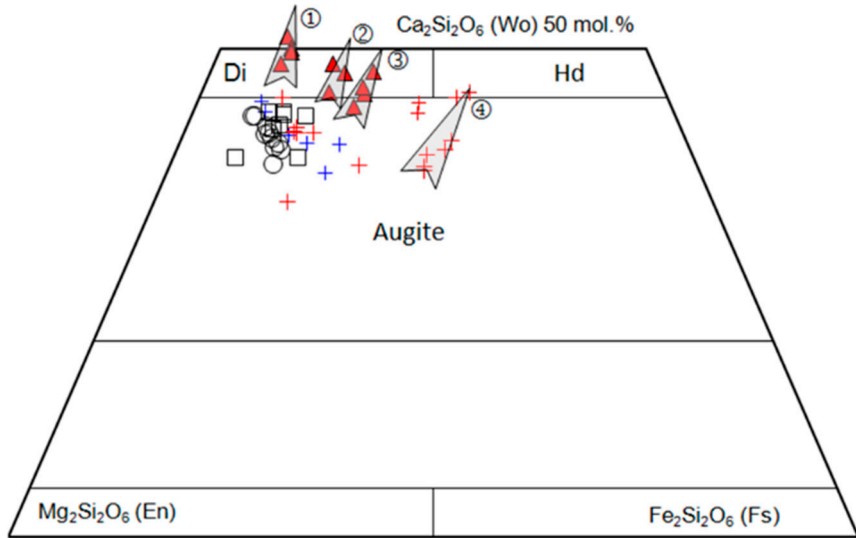

**Figure 5.** Chemical composition of "QUAD" Cpx from basaltic eruptive rocks in the system Wo-En-Fs according to the classification diagram of [69]. Symbols: squares = sample Les-1, circles = sample RD2/17, red triangles = sample Les-3, blue crosses = sample Les-1A (zones with An-rich Pl), red crosses = Les-1A (micropoikilitic zones with Ab-rich Pl). Arrow trends ①–④ show an increase in the Wo component in Cpx due to CaCO$_3$ assimilation.

The development of reverse trends from the main Px fractionation line Di → Fs is the result of an increase in calcium activity in the differentiating interstitial melt. Each of the identified fractionation branches indicates a local change in calcium activity because of an externally generated change in the solidifying system. The main candidate is the dissolution of the incorporated carbonate fragments. Dissolution and assimilation of carbonates and its effect on the development of the solidification system have been analyzed experimentally [70–73], but not at pressures corresponding to shallow subsurface conditions (e.g., near 1 Atm), although this has been documented in some field studies (e.g., paralava formation from Eastern Mongolia; [74]). Assimilation of carbonate material leads to an increase in calcium activity in melt, a shift in the composition of the assimilating magma to alkaline type [70–72], and an increase in magma oxidation conditions [73,75], which corresponds to the observed increase in Wo and $Fe^{3+}/Fe^{2+}$ in crystallizing Cpx.

The calculated $Fe^{3+}/Fe_T$ in Cpx allows the application of Cpx oxybarometry for Cpx with low Na contents [76] to obtain magmatic redox conditions as follows: basalt Les-3 = ΔQFM of 2.2 ± 1.3 ($n$ = 11), dolerite RD2/17 = ΔQFM of 0.9 ± 0.8 ($n$ = 10), and dolerite Les-1A2 = ΔQFM of 0.1 ± 0.8 ($n$ = 13). The progressive increase in $Fe^{3+}/Fe^{2+}$ in pyroxene with respect to Al/Ti differentiation confirms the change in redox conditions during the solidification of the individual igneous bodies (Figure 6).

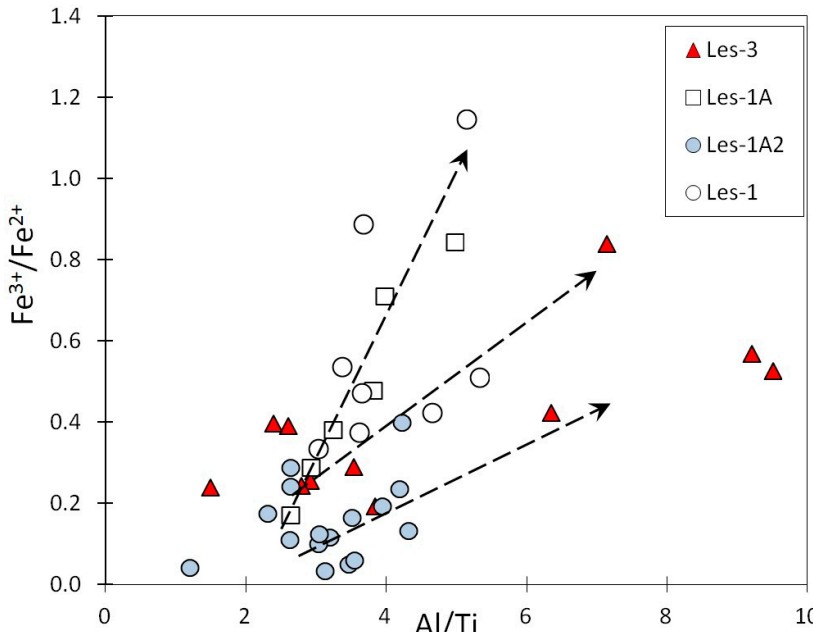

**Figure 6.** $Fe^{3+}/Fe^{2+}$ vs. Al/Ti ratio, reflecting the differentiation state of basaltic magma. $Fe^{3+}$ is calculated using the charge balance within the Cpx lattice. Different arrow trends indicate diverse and dynamic redox conditions within the separate subvolcanic and effusive bodies.

### 3.2.2. Plagioclase

The composition of Pl is very distinctive in each of the analyzed samples. Albite is a common rock-forming mineral in basic volcanic and intrusive rocks, but its presence in basalts is generally due to subsolidus reactions with a percolating fluid phase during final solidification and post-magmatic processes [77,78]. Albite occurs in all five samples. The magmatic origin of albitic Pl was confirmed from Pl cores without inclusions as having originated by subsolidus alteration reactions. Importantly, prehnite (Prh), Ep, zoisite (Zo), and clay minerals, a common mineral assemblage in Ab and linked to the saussuritization of Ca-rich feldspars (Fsp), were not recognized. The habit of the Pl crystals is columnar to acicular and with a transition to skeletal crystal forms that are an expression of a purely magmatic process controlled by magma undercooling and degassing.

The relationship between Fe and Mg (Fe+Mg (mol.%) and Fe/Mg ratio) as a magmatic fractionation index changes depending on the Ab and or orthoclase (Or) contents, where an increase in Ab % is accompanied by a quantitative decrease in Fe+Mg together with an Fe/Mg increase. In the case of Or %, Fe+Mg and Or contents rise with a Fe/Mg decrease. These characteristics support the primary magmatic origin of albitic Pl. Based on the systematic presence of Fe and Mg, our data are recalculated to chemical formulas on the basis of five cations and Ab-anorthite (An)-Or solid solution with Fe+Mg (Table S1).

Subvolcanic dolerite Les-1A contains two separate populations of Pl. The first one is the product of magmatic crystallization with a composition $An_{62.4-51.3}Ab_{44.1-32.9}Or_{4.4-0.6}MgFe_{5.2-3.8}$, with Mg+Fe correlating positively with Or % (r = +0.93) and Fe/Mg with Ab % (r = +0.93). Reconstruction of Pl compositions using simulations of intruded basaltic melt (composition Les-1 after $CaCO_3$ correction) using rhyolite-MELTS for different ($\Delta QFM = 0$, $\Delta QFM = +1$, $\Delta QFM = +2$, and $\Delta QFM = +3$) redox conditions, for equilibrium and for fractional crystallization, achieved 62–51 An mol.% composition in the temperature range of 1160–1110 °C. The lower temperature value (1110 °C) corresponds to the observed Pl composition of 51% An, which was achieved in the simulation at 50%–60% solidification volume, at which stage the Pl compositional evolution ends.

The Pl in the other basalt samples is Ab but still corresponds to Fsp in the An-Ab-Or ternary system or the An-Ab-Or-MgFe system ($An_{5.4-2.6}Ab_{94.8-85.5}Or_{1.1-0.3}MgFe_{8.0-0.4}$). As the Ab content increases with Fe/Mg (r = +0.8), the Mg+Fe total decreases (r =−0.97). Such a Na-rich Pl composition is observed in dolerite sample Les-2 ($An_{4.7-0.1}Ab_{99.2-92.7}Or_{3.3-0.2}MgFe_{1.7-0.5}$) with the increase in Ab accompanied by an increase in Fe/Mg (r = + 0.94). Plagioclase in basaltic lava Les-3 is ternary Ab containing Mg and Fe ($An_{2.9-0.5}Ab_{98.7-93.1}Or_{0.7-0.2}MgFe_{4.7-0.5}$), and a negative relationship between Mg+Fe and Ab is seen (r =−0.84). Other correlations are not observed, probably due to the disturbed crystallization kinetics under physically open solidification conditions after lava eruption. The feldspar chemistry is probably a result of the simultaneous cotectic Pl-Cpx crystallization.

Each of the albitic Pl populations contains Fe and Mg despite the different chemical differentiation trends and different solidification histories. A general similarity in Pl compositions is demonstrated in the diagrams Or vs. (Mg+Fe) and Or vs. (Fe/Mg) (Figure 7A,B), with Mg+Fe and Fe/Mg controlled by igneous differentiation. Ca-rich Pl (sample Les-1A with An 62–51 mol%) shows different variations between Or and Fe/Mg (Figure 7A,B) in comparison to albitic Pl from dolerite Les-2. The higher Mg+Fe content of the Ca-rich Pl indicates a more primitive character of the liquid. The albitic Pl from basaltic andesite lava Les-3 shows a different Mg+Fe and Fe/Mg evolution relative to the subvolcanic basalt bodies because of open system lava degassing (Figure 7A,B).

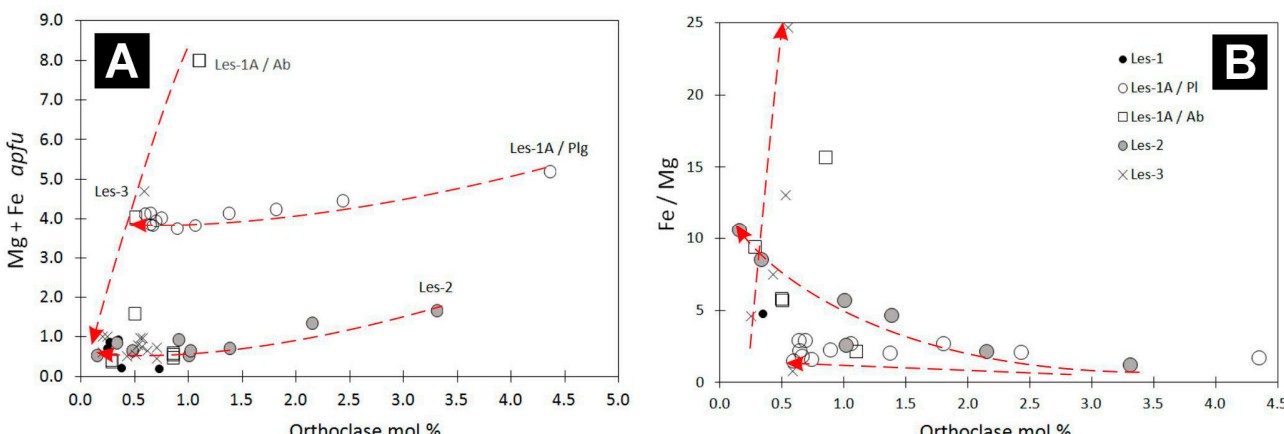

**Figure 7.** (**A**) Evolution of Or content (mol.%) in Pl depending on Σ Mg+Fe and Fe/Mg (**B**). Both, vertical and horizontal trends of Pl chemical composition are controlled by melt differentiation. Decrease in compatible Fe+Mg and increase in Fe/Mg is observed and shown by arrows for different samples.

Despite the direct effect of supercooling on the feldspar skeletal habit of potentially albitic Pl, as experimentally demonstrated [79], or an effect of Mg+Fe on Pl structure (Figure 7A) [80–82], primary igneous albitic Pl in basic eruptive rocks was not commonly reported [83]. Thus, syn- and post-eruptive magma undercooling or extensive fractionation of Cpx may fail to generate a feldspar Ab composition. Fractionation of Aug and Di fails to explain the abrupt change in Pl composition, while the small magma volumes do not have the thermal potential to make the subvolcanic system capable of differentiation to reach a residual haplogranite liquid composition. The common low-thermal subsolidus Ttn-Act-Chl-Cal ± Ep assemblage and the Cal-filled amygdales could indicate a high calcium activity during final solidification but do not explain the lack of Ab in this assemblage. Therefore, the widespread presence of albitic Pl in different eruptive facies (feeder dykes/lavas), various habits (tabular, acicular, skeletal), and various textures (ophitic, granophyric, subophitic, microphyric) calls for a principal controlling mechanism that maintains Ab compositional stability during the solidification of fractioning basalt magma.

### 3.2.3. Fe-Ti Oxides

Fe-Ti oxides in the studied rocks are comprised of TiMag s.s. with trace contents of Al (0–0.002 *apfu*) and spinel s.l. with Al contents between 0.003 and 0.014 *apfu*. Recalculated chemical formulas are given in Table S1. The morphology of Fe-Ti crystals was controlled by non-equilibrium crystallization influenced by magma undercooling and devolatization in low-pressure subsurface conditions. The morphology of tiny isometric crystals often shows skeletal to dendritic shapes (Figure 3A,C). Titanomagnetite crystals are either pure or contain very thin irregular exsolutions of $SiO_2$ (sample Les-1). The TiMag in each sample exhibits variable ulvöspinel (Uspl) contents: Les-1: 71.7-61.3-44.3 mol%; Les-1A: 58.4-35.3-28.5 mol%; Les-3: 42.8-14.7-2.0 mol% (Figure 8). Fe-Ti oxides or spinels with higher aluminum were identified in sample Les-1A with relics of Ca-rich Pl. Samples with simple albitic Pl have low Al. Fe-Ti oxides in the Chl-Act-Ttn ± Cal mineral assemblage are pure Mag and are locally associated with Hem (identified by EDS spectra). These magnetites and hematites are the results of melt oxidation by free $CO_2$ bubbles captured in a crystal mush of low permeability at advanced (up to 50 vol.%) magma solidification.

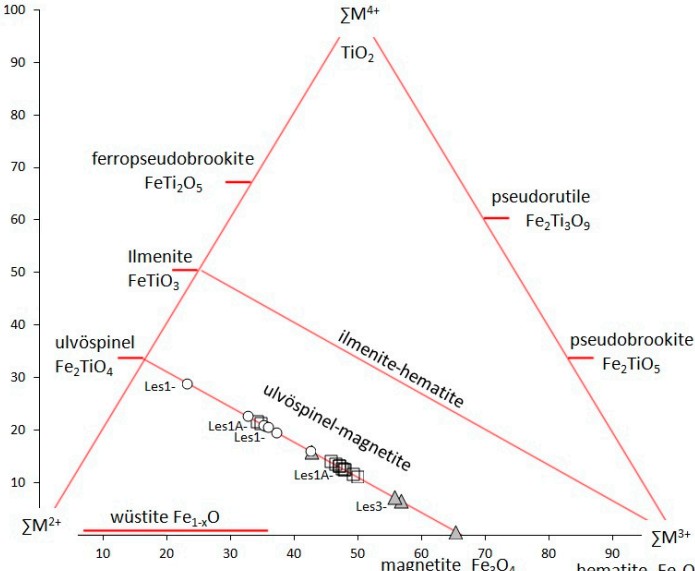

**Figure 8.** The chemical composition of Fe-Ti oxides in the sample suite. Fe-Ti oxides correspond to titanomagnetite with a solid solution between Uspl and Mag. The projected chemical composition employs the recalculated crystal formula based on three cations using the program ILMAT [84]. Symbols: squares = Les-1A; circles = Les-1; triangles = Les-3.

### 3.3. Classification of Analyzed Rocks

The subvolcanic and volcanic rocks are classified by field, petrographic, and chemical criteria. Due to the similar geochemical and isotopic characteristics of basaltic lavas and subvolcanic dolerites, the subvolcanic bodies likely represent the feeder channels for the basaltic lavas into an aquatic environment, as indicated by the presence of hyaloclastites. The ophitic, subophitic, and intersertal doleritic textures are typical of solidification in a subvolcanic environment. Ol?-Cpx-Pl-TiMag and, more often, Cpx-Pl-TiMag are identified as primary magmatic mineral assemblages. Magmatic minerals are spatially closely associated with a subsolidus Chl-Ab-Cal-Act-Ttn-Ep assemblage characteristic of alteration processes that modify the primary composition of volcanic rocks (e.g., spilite and keratophyre; [85]). The presence of amygdales and incorporated limestone fragments indicate a chemically open system during eruption with syn-eruption assimilation. Assimilation of carbonate and syn-eruption degassing modified the sample chemical composition ($CaCO_3$ addition accompanied by loss of $CO_2$, $SO_x$, $H_2O$). Both processes operated together along with classical magmatic differentiation in subvolcanic feeders. The effect of alteration on the chemical composition of the samples is also shown by the loss on ignition (LOI) values of 7.7–4.6 wt.% and $CO_2$ 5.47–1.24 wt.% ($CO_2$ recalculated from mol equivalent TOT/C wt.%).

On the other hand, the chemically conservative nature of subsolidus reactions is deduced for non-volatile elements by petrographic observations, where subsolidus mineral associations fill the interstitial space of homogeneous magmatic textures and are characteristic of synvolcanic auto alteration. For whole rock analyses with high LOI, they are recalculated on an anhydrous basis. Chemical analyses are independently recalculated to remove assimilated $CaCO_3$ as a mol equivalent of TOT/C (wt.%). Chemical data and their recalculated equivalents are presented in Table S1.

The total alkali-silica (TAS) classification scheme (Figure 9) for volcanic rocks [86] identifies two parallel differentiation trends in the sample set, which are specified by the discriminant line of [87] for alkaline and subalkaline types. According to the TAS IUGS classification [86] and the criteria for alkaline/subalkaline volcanic rocks [87], the rocks correspond to sub-alkaline basalts (Les-5, RD2/17), sub-alkaline basaltic andesite (Les-3) on the "SA" trend and alkaline basalt (Les-1), alkaline trachybasalt (Les-1A) to alkaline basaltic trachyandesite (Les-2) on the "A" trend. Some samples retain their "A" (Les-1, Les-1A, Les-2) or "SA" (Les-5, Le-3, RD2/17) characteristics even after correction for $CaCO_3$ or change in favor of "SA" (RD5-07). The lavas (Les-3, Les-5) maintain a subalkaline character even after correction for $CaCO_3$. The calculation of the CIPW normative composition of the samples shows a presence of normative nepheline (Nph) 2.31–0.56 wt.% in the alkaline types, while in subalkaline samples, it is absent. Normative quartz (Qz) as a criterion for subalkaline Qz-normative rocks was identified only in the case of RD-5/07 dolerite. Using the criteria for distinguishing the subalkaline tholeiite and calc-alkaline differentiation series ($SiO_2 = 6.4 \times FeO/MgO + 42.8$; [88]), it is shown that some subalkaline samples have a calc-alkaline character, but some samples after $CaCO_3$ correction belong to the tholeiite differentiation series (Les-3, Les-5, RD2/17). When comparing sample projection with and without $CaCO_3$ correction, the effect of $CaCO_3$ is significant. Following $CaCO_3$ correction, some basalts move from alkaline to subalkaline or from calc-alkaline to tholeiitic composition depending on the modal abundance of Cal but broadly maintain their fields on classification diagrams. The potential effect of open alteration on the chemical composition and classification of basalts is controlled by immobile element ratios Zr/Ti, Nb/Y [1], which assign all samples to the subalkaline basalt and andesite fields (not shown) and broadly supports the TAS classification results. Despite the influence of $CaCO_3$ on rock classification, the sample suite clearly represents sub-alkaline basalts to basaltic andesites and alkaline basalts to trachybasalts. Basaltic rocks are generally metaluminous because aluminum/alkali indices based on molar ratios are ACNK (0.4–0.63) < 1 and ANK (1.53–2.25) > 1, respectively, for $CaCO_3$ corrected whole-rock ACNK data (0.55–0.77) < 1.

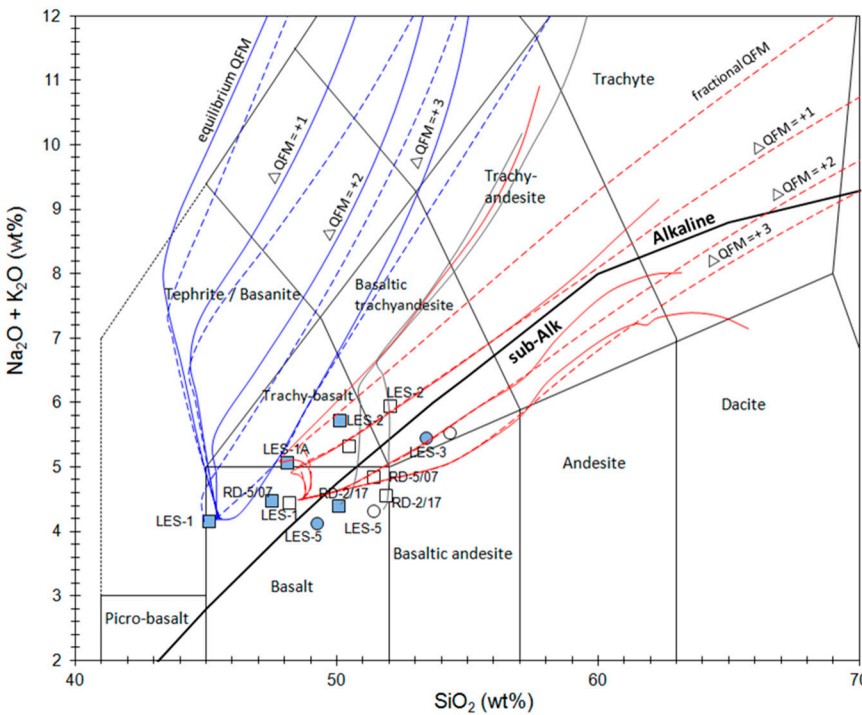

**Figure 9.** TAS (total alkali-silica) classification of volcanic rocks [86]. The samples are converted to a 100% anhydrous basis (solid symbols) and then employ $CaCO_3$ correction using the molar equivalent of the analyzed TOT/C (open symbols). The discriminant lines for alkaline "A" and sub-alkaline "SA" rocks are from [87]. The composition of the samples corresponds to two trends: alkaline basalts to trachybasalts, and subalkaline basalts to basaltic andesites. Model simulations after $CaCO_3$ correction (red lines) and without $CaCO_3$ correction (blue lines) are also shown. Magmatic differentiation was simulated using rhyolite-MELTS v.1.2.0 [63–65] as the evolution of residual liquid under near-surface conditions ($p$ = 10 bar), equilibrium (solid lines), and fractional crystallization (dashed lines) for different fixed redox conditions QFM and $\Delta$QFM = +1; +2, +3 and $H_2O+CO_2$ saturation. Values for discriminant lines are presented in Table S1.

### 3.4. Geochemistry and Inferred Melted Mantle Sources

Measured and recalculated chemical analyses are presented in Table S1 (whole-rock analyses). Analyses with high LOI and TOT/C values were recalculated by correcting for the molar equivalent of $CaCO_3$. Despite the petrographic identification of Cal in the form of lithic fragments ("xenoliths") and amygdaloidal structures, only part of the total calcium currently bound to $CaCO_3$ is of external origin as there is redistribution of primary magmatic calcium between magmatic liquid, fractionating phases, and amygdaloidal structures. Analyses without CaO removal are thus also presented and range from 14.36 to 6.68 wt.%; following removal using the normative TOT/C equivalent $CaCO_3$, the range is 12.78–5.45 wt.%. Other element totals are also affected. The effect of $CaCO_3$ reduction on the whole rock analyses is shown in Figure 10. The $CaCO_3$ correction improves the poor correlation between CaO vs. the differentiation $Al_2O_3/TiO_2$ ratio to a direct linear fractionation trend with $r^2$ = 0.94 (Figure 10). $Al_2O_3/TiO_2$ also negatively correlates with the fractionation of incompatible Nd ($r^2$ = 0.85) or $Na_2O$.

Based on the $SiO_2$-$Na_2O+K_2O$ "TAS" classification [86], the basaltic volcanic rocks are divided into alkaline and subalkaline trends (Figure 9), and magmatic fractionation differentiates basalts from subalkaline basalts to basaltic andesites and from alkaline basalts and trachybasalts to basaltic trachyandesites. Both identified trends branch from a common starting point, which represents the composition of magma corresponding to sample Les-1, from which it is possible to derive the compositions of alkaline and sub-alkaline trends, as shown by rhyolite-MELTS model reconstructions for various redox conditions and fractionation paths (Figure 9). The chemical composition of the rocks reflects branched

fractionation trends, as shown by the rhyolite-MELTS simulations for different redox conditions $\Delta QFM = 0, +1, +2,$ and $+3$. $SiO_2$ as the differentiation factor shows very poor correlations with incompatible REEs ($r^2$ values of 0.03–0.43). In contrast, differentiation relationships between MgO (8.60–6.14 wt.%) and Ni (244–105 ppm) yield very high $r^2$ values (+0.99).

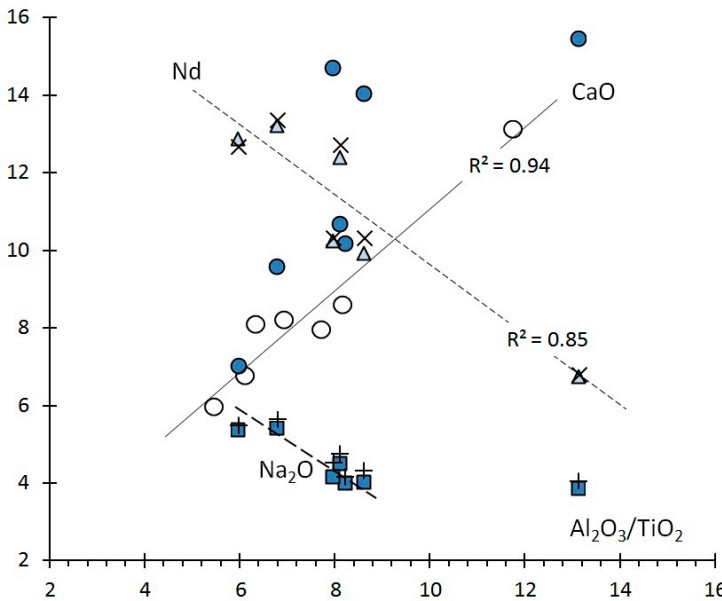

**Figure 10.** The effect of $CaCO_3$ removal on the whole rock chemical data. $Al_2O_3/TiO_2$ is plotted separately for CaO (wt.%), $Na_2O$ (wt.%), and Nd (ppm). Original data are filled symbols; corrected data are open symbols. The corrected CaO data correspond to a linear differentiation trend ($r^2 = 0.94$). Full blue symbols for CaO ●, $Na_2O$ ■, Nd ▲ are original whole rock data (in wt.% or ppm), and next symbols for CaO ○, $Na_2O$ +, Nd × correspond to their $CaCO_3$ reduced equivalents.

While $SiO_2$ and the REE contents do not show evidence of fractionation, $Na_2O$ and CaO after correction show a direct fractionation relationship, which is caused by magmatic differentiation with an increase in incompatible $Na_2O$ (4.06 to 5.64 wt.%) and a decrease in CaO (11.74 to 5.45 wt.%) after phenocryst separation. The fractionation relationships of $Na_2O$ and CaO also show correlations with incompatible REEs where $r(Ca-REE) \leq -0.90$ and $r (Na-REE) = +0.73$ to $+0.93$. The REE systematics of the basalts are very specific. With respect to the broadly coeval Miocene volcanism of the Western Carpathians, which have a calc-alkaline differentiation trend with $La/Sm_n > 1$ and $Sm/Yb_n \geq 1$ [89–91], the REE systematics of the Lesnica Valley basalts are characterized by an "N-MORB" pattern with both $La_N/Yb_N$ and $La_N/Sm_N < 1$ at nearly constant $Eu/Eu^*$ values (~0.9) (Figure 11). However, they do differ in other REE ratios ($Sm/Yb_n$ and $Gd/Yb_n \geq 1$) from N-MORB and OIB-like rocks ($La/Sm_n < 1$ and $Sm/Yb_n < 1$).

Models of Spl and Grt mantle peridotite melting (Figure 11) imply 1% melting of Grt peridotites and 7%–8% melting of Spl peridotites mixed in a mass ratio of L (GrtP):L (SplP) 1:6 to 1:11, which generate the observed REE ratios ($La/Sm_n < 1$ and $Sm/Yb_n > 1$). The presence of negative Eu-anomalies and low temperatures for the magma liquidus (~1200 °C) indicate a previous phase of fractional crystallization prior to intrusion. The decrease in Ni concentrations (271–248 to 125–105 ppm) at constant $Eu/Eu^*$ values suggests that fractionation under near-surface conditions of a shallow dyke plumbing system does not generate Eu-anomalies, despite ideal differentiation (CaO decrease with $Na_2O$ and REE increase).

The LILE–HFSE–REE systematics on an N-MORB-normalized multi-element diagram (Figure 12) show significant LILE (Rb, Cs, Ba, K) and Pb enrichment. The HFSE are quantitatively linked to magma fractionation, where an increase in Nb (2.6 to 5.9 ppm) and Zr (81.4 to 157 ppm) is accompanied by an increase in $Na_2O$, REE and a decrease in CaO; $r(Nb-Nd) = +0.97$; $r(Zr-Nd) = +0.97$ and $r(Na-Zr) = -0.95$. The REE–HFSE relationship

shows a weak negative Nb anomaly and systematically slightly positive Zr anomalies, as shown by $Nb/La_n$ = 1.07–0.83, $Zr/Sm_n$ = 1.19–1.15, or in the expression $\Delta Zr$ = +0.06 to +0.19 (according to [92]). The HFSE/REE ratios (Nb/La, Nb/Sm, Zr/Sm, Zr/Yb) remain constant during fractionation of $Na_2O$, $K_2O$, $Al_2O_3/TiO_2$.

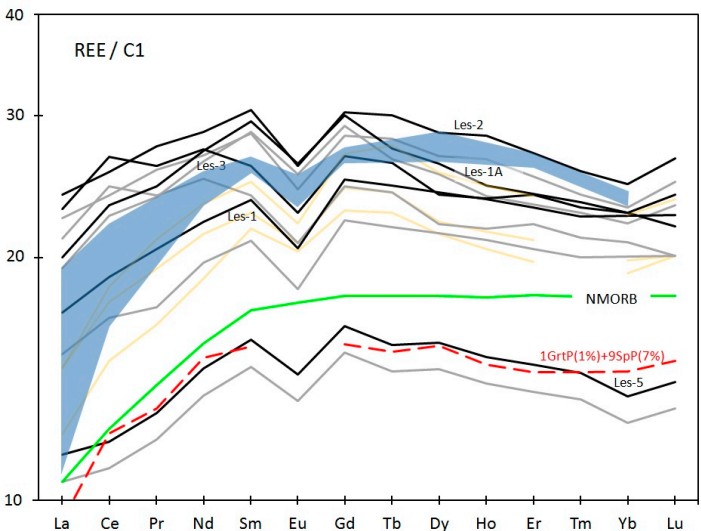

**Figure 11.** Chondrite-normalized REE distribution of basic volcanic rocks from the Lesnica Valley. Chondritic C1 normalization values are from [86], with measured REE (grey solid lines) and REE after $CaCO_3$ removal (black solid lines) highlighted. N-MORB [93] is shown for comparison (green). The dashed lines represent models of fractional partial melting of mantle peridotites with a primordial mantle composition (data from [93]). Spinel peridotite 0.578Ol + 0.27Opx + 0.119Cpx + 0.033Spl (in wt.%) is melted according to the peritectic reaction 1L + 0.167Ol = 0.652Opx + 0.466Cpx + 0.049Spl. Grt peridotite 0.598Ol + 0.211Opx + 0.076Cpx + 0.115Grt (in wt.%) is melted according to the peritectic reaction 1L + 0.152Opx = 0.044Ol + 0.964Cpx + 0.138Grt. Peritectic reactions, composition of mantle peridotite, and distribution coefficients are taken from [94]. The shaded blue field is constructed from the Px-Pl-Grt-Spl "granulite" xenoliths from the Western Pannonian basin alkaline basalts [95]. Basalts from the potential analog of the Blanco transform fault system are indicated by yellow curves [96].

Unlike the REE, HFSE, $Na_2O$, and CaO, the LILE relationships cannot be explained by fractionation as r(LILE–REE), r(LILE–HFSE), r(LILE–Na) and r(LILE–$CaCO_3$) are close to 0, which can only be explained by variable external input of LILE. In particular, the lack of correlation between $Na_2O$ and LILE indicates that LILE enrichment is not the result of mantle source processes or magma interactions with the continental crust, as LILE entry into the magmatic system would lead to synchronous fractionation of Na and LILE in subvolcanic feeder channels. However, internal relationships between LILEs are strongly correlated, with r(K-Rb) = +0.91, r(K-Cs) = +0.97, r(K-Rb) = +0.98, but with no correlation between Sr and LILE. Positive Pb anomalies, whose distribution is connected to carbonate assimilation r($CaCO_3$-Pb) = +0.72, behave differently to strontium r($CaCO_3$-Sr) = −0.88, probably because of AFC. The LILE–REE–HFSE systematics can be understood as the result of advanced 7%–8% melting (low $La/Sm_n$, low $La/Yb_n$, $Sm/Yb_n \geq 1$) of mantle peridotites without significant previous modification of the mantle source ($Nb/La_n$ and $Zr/Sm_n$ = ±1) and enrichment by LILE during syn-eruptive contamination.

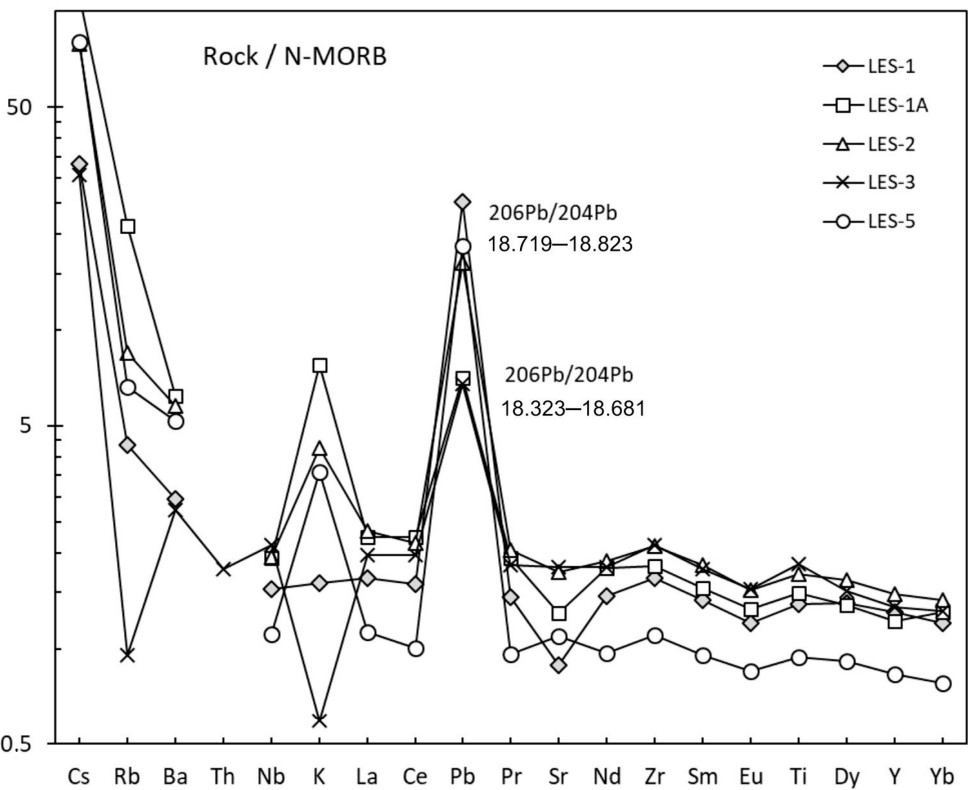

**Figure 12.** N-MORB-normalized [93] multi-element diagram of the basaltic rocks from the Lesnica Valley. Chemical analyses are recalculated following $CaCO_3$ removal.

### 3.4.1. Sr, Nd, and Pb Isotope Compositions and Comparison with the CPR

Whole-rock Sr, Nd, and Pb isotope analyses are also presented in Table S1. Syn-eruptive contamination is particularly pronounced for Pb and Sr, which are reflected in significant positive Pb anomalies on multi-element N-MORB normalized patterns and variable Sr (Figure 12), with r(Pb to $CaCO_3$) up to 0.7 and r(Pb to LOI) as a potential indicator of alteration or contamination up to 0.84. In Sr, Pb, or Nd isotopic space, for simple magmatic fractionation, radiogenic isotope ratios are expected to be independent of element concentrations. The opposite is observed. The correlation of Nd and Sm concentrations with isotope ratios is r(Nd to $^{143}Nd/^{144}Nd$) = +0.82 and r(Sm to $^{143}Nd/^{144}Nd$) = +0.78, which after $CaCO_3$ correction increases to r (Nd to $^{143}Nd/^{144}Nd$) = +0.87. Relationships of Pb concentration with the isotope ratio $^{206}Pb/^{204}Pb$, as a sensitive indicator for assimilation and contamination by crustal radiogenic material, yields a significant value of r(Pb to $^{206}Pb/^{204}Pb$) = +0.67. Thus, the Pb isotopic composition likely does not represent the original magma composition.

Initial $^{87}Sr/^{86}Sr$ isotope ratios of basaltic rocks from the Lesnica Valley after correction for an eruption age of 13 Ma are scattered in a wide range of 0.70705–0.70584 but with a rather narrow variation of $^{143}Nd/^{144}Nd$ 0.513025–0.512991, or $\varepsilon Nd_{(13)}$ = +8 to +7.4 (CHUR = 0.512638; [7]). Exceptionally, sample Les-5 (basaltic lava) with a hyaloclastite texture has a lower $\varepsilon Nd_{(13)}$ = +3.0. Considering petrographic, petrological, and geochemical results for syn-eruptive contamination of the basaltic magma, the five analyzed samples from separate igneous bodies represent different levels of contamination. The $^{87}Sr/^{86}Sr$ ratios show a systematic increase towards higher radiogenic values typical for contamination by crustal material. The influence of carbonate assimilation on the Sr-Nd isotopic composition of basaltic rocks is demonstrated by mixing line construction between WPB "granulite" xenoliths described below and theoretical upper crustal sediments [20] with high Sr/Nd typical of carbonates (Figure 13). The UC-WPB xenoliths mixing line shows that with limited assimilation of UC sediments (up to 5–10 wt.%) by basaltic magmas

from Lesnica, $^{143}Nd/^{144}Nd$ remains close to primary compositions and maintains their primitive $\varepsilon Nd$ signature, while $^{87}Sr/^{86}Sr$ increases significantly. The narrow variation in $^{143}Nd/^{144}Nd$ ratios for four samples (Les-1, Les-1A, Les-2, Les-3) ($\varepsilon Nd_{(13)}$ = +8 to +7.4) is typical for a significantly depleted mantle source. The high positive $\varepsilon Nd$ values and narrow variability of $^{143}Nd/^{144}Nd$ suggest that the Nd isotope values can be considered close to being primary for intruding basaltic magma and, thus, can be used for temporal and regional geochemical correlations of the Cenozoic volcanism in the Carpathian–Pannonian region (CPR) (Figure 13).

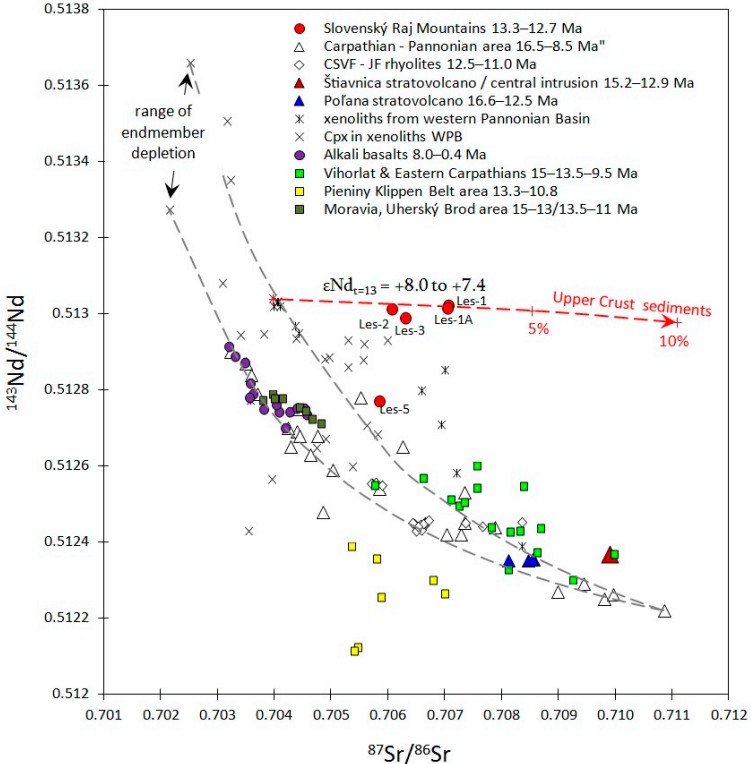

**Figure 13.** Isotopic composition of Neogene and Quaternary volcanic rocks from the CPR region. The $^{87}Sr/^{86}Sr$ and $^{143}Nd/^{144}Nd$ isotope ratios are corrected for eruption age. The dashed lines are endmember mixing lines illustrating the isotope evolution of the magmatic sources in individual CPR volcanic provinces. The dashed arrow shows the effect of $^{87}Sr/^{86}Sr$ contamination on the SRM volcanic rocks. Sources of $^{87}Sr/^{86}Sr$ and $^{143}Nd/^{144}Nd$ isotope data from the CPR are summarized in [14,16–22,24,25,27,29,30,35–38,40,43,44,90,91,95,97–107].

The $\varepsilon Nd$ values of the SRM basalts are even higher than those of primitive alkaline basalts from the CPR [19,20,108]. SRM basalts have very similar $\varepsilon Nd$ values to "granulite" xenoliths (+6.4 to +8.1; [95]) from the alkaline basalts of the Western Pannonian Basin (WPB, Figure 13). These mafic xenoliths, composed of Cpx, Pl, Opx, Pl, spinel (Spl), olivine (Ol) $\pm$ garnet (Grt), Amp, and Ttn, have compositions similar to magmatic cumulates from parental MORB-like Ol normative tholeiitic magmas [95,109] and a magmatic origin has been proposed. The similarity between these mafic xenoliths and the basaltic rocks from SRM is also supported by the depleted LREE pattern and the expected tholeiite character of magma (Figure 11). These similar geochemical characteristics most likely indicate a similar composition of the mantle source. The chemical composition of the LREE-depleted SRM rocks with a markedly depleted isotopic composition $\varepsilon Nd$ > +7.3, compositionally similar to oceanic basalts, is distinct and unusual for rocks from other Cenozoic and Quaternary volcanic CPR provinces.

Although primary Sr isotopic compositions cannot be determined for the SRM basalts due to carbonate assimilation, the Nd isotope values are likely close to primary signatures.

A projection in Sr-Nd isotope space (Figure 13) with variable Sr contamination likely implies a composition similar to the mafic "granulite" xenoliths from the WPB and, thus, a more depleted mantle source for the SRM magmas than for the alkaline CPR basalts (another trend on Figure 13). The SRM basalts and the WPB "granulite" xenoliths isotopic systematics show that the melted peridotite source was isolated from refertilizing percolating melts from the lower mantle. Such a situation is typical of the melting of an isolated region of the mantle, such as supra-subduction lithospheric mantle, where melting is controlled by external agents such as fluid metasomatism and tectonic stresses associated with subduction.

This contrasts with the petrogenesis of the high K alkaline volcanics of the volcanic provinces in the OWC (Moravia and PKB), where thermal erosion and interaction of the lithosphere with rising asthenosphere melts have been postulated [27,30]. The small volume of erupted magmas in the Lesnica Valley and the short emplacement time interval (13.3–12.7 Ma) may also support an isolated depleted mantle source during a restricted tectono-magmatic pulse.

### 3.4.2. MELTS Simulations

The evolution of REDOX conditions based on the calculated $Fe^{3+}/Fe^{2+}$ of analyzed Cpx yields values between $\Delta QFM = 0$ and $\Delta QFM = +2$, which correspond to the reconstructed fractionation trends of "liquid lines of descent". During the solidification of basaltic magma in a shallow environment, fluid phase saturation and contemporaneous degassing are expected since $H_2O$ and $CO_2$ solubility is very limited at low pressure [110]. The control of the saturated dissolved $H_2O+CO_2$ fluid in rhyolite-MELTS simulations, therefore, requires low pressure. Prior to each MELTS simulation, the silicate whole rock analysis was converted to a 100% anhydrous basis. A 1:1 mixture of $H_2O$ and $CO_2$ was used as an input value before the calculations. Subsequently, a model amount of 1 wt.% $H_2O$ and 1 wt.% $CO_2$ was inserted and re-normalized to 100%. The measured iron content ($Fe_2O_3$ total) was recalculated for the model conditions $\Delta QFM = 0, +1, +2, +3$ by equilibration at a temperature of 1300 °C. The chemical composition of the liquid at 1300 °C, supplemented with saturated $CO_2$ and $H_2O$ with modified $Fe^{3+}/Fe^{2+}$, was subsequently used to simulate equilibrium and fractional crystallization to reconstruct the phases, volume, and degassing evolution during solidification of the basaltic melts. The resulting rhyolite-MELTS simulations are given in Table S1, and the comparison of chemical and physical characteristics between samples is given in Table 1.

**Table 1.** Simulations of the chemical variability of the analyzed samples. The samples lie on a liquid line of descent and differentiate from the parent magma Les-1. The model redox conditions and magma temperature (°C), $\Delta SiO_2$ and $\Delta Alk$ (in wt.%), and the degree of differentiation (vol.%) are presented. Samples with a closer composition to simulated magma compositions are highlighted in grey.

| $CaCO_3$ Correction | Sample | REDOX | $SiO_2$ wt.% | $Na_2O+K_2O$ wt.% |
|---|---|---|---|---|
| | | $\Delta QFM$ | T (°C)-$\Delta SiO_2$-Diff (vol.%) | T (°C)-$\Delta Alk$-Diff (vol.%) |
| Ca-uncorrected | Les-1A | 0 | 1070-(0.1635)-69.4 | 1140-(0.061)-29.4 |
| Ca-corrected | Les-1A | +1 | 1130-(0.274)-36.9 | 1130-(0.041)-36.9 |
| Ca-uncorrected | Les-2 | 0 | 1130-(0.434)-34.4 | 1110-(0.118)-48.3 |
| Ca-corrected | Les-2 | +1 | 1120-(1.648)-48.5 | 1120-(0.37)-48.5 |
| Ca-uncorrected | RD2-17 | +3 | 1200-(0.707)-4.99 | out |
| Ca-corrected | RD5-07 | +2 | 1160-(0.426)-17.13 | 1160-(0.074)-17.13 |
| Ca-uncorrected | Les-3 | +2 | 1150-(0.019)-26.4 | 1150-(0.107)-26.4 |
| Ca-corrected | Les-3 | +2 | 1150-(1.021)-26.4 | 1140-(0.062)-36.8 |

The dolerite bodies and effusive rocks have their own specific chemical features, which can be derived from the different degrees of magmatic fractionation of one parental melt. As there is unlikely a physical fractionation connection between the bodies, the chemical differences between the dolerite bodies are the result of pre-intrusive processes and prior to syn-intrusive contamination by limestone. The samples show two differentiation magmatic

trends (alkaline and subalkaline; Figure 9). The alkaline differentiation trend can be explained as a product of subalkaline tholeiitic magma fractionation modified by carbonate rock assimilation (Les-1, Les-1A, Les-2). The subalkaline differentiation trend represents subalkaline magma fractionation with a low degree of carbonate assimilation with little modification toward calc-alkaline differentiation on the "liquid line of descent" (Les-5, Les-3, RD-5/07, RD-2/17).

Simulations of differentiation pathways (Figure 9) from a primitive basaltic magma with the composition of Les-1 (after $CaCO_3$ removal) at different $\Delta$QFM can simulate the chemical variability of the analyzed dolerites and lavas from the volcanic-intrusive complex. Significant fractionation trends appeared between the dolerite Les-1 (after $CaCO_3$ correction) and samples Les-1A, Les-2, RD5-07, Les-3 (after $CaCO_3$ correction) or the basalt Les-3 without $CaCO_3$ correction. The magmatic differentiation simulations are summarized in Table 1, which lists the relevant temperatures (°C) and the degree of differentiation (vol.%). The degree of differentiation is identified based on the minima $f(SiO_2)$: minimum $|\Delta SiO_2|$ and $f(Alk)$: minimum $|\Delta Alk|$. $\Delta SiO_2 = SiO_{2\text{-analyzed}} - SiO_{2\text{-calculated}}$. $\Delta Alk = Alk_{\text{-analyzed}} - Alk_{\text{-calculated}}$. $Alk = Na_2O + K_2O$.

Analysis of the $\Delta SiO_2$ and $\Delta Alk$ minima identified the corresponding values of magmatic differentiation degree and interstitial residual melt temperatures. The calculated values of differentiated magma batches (in vol.%) are used for reconstructing the parental magma reservoir at the time of magma extraction and eruption. All calculated degrees of differentiations are less than 50 vol.%–60 vol.%, which represent the limits for dyke propagation and magma eruption [111–114]. The calculated degrees of differentiation indicate that the basaltic magmas were modified by fractional crystallization in a shallow magmatic reservoir, from which they were evacuated and intruded into the country rock in pulses. Different levels of differentiation indicate the pulsating nature of eruptions. The volcanic rock chemistry was controlled by the rheology of stored magma, which is linked to its solidification history (Figure 14), with the interstitial melts remaining trapped in a crystal mush that operated as a mechanical barrier.

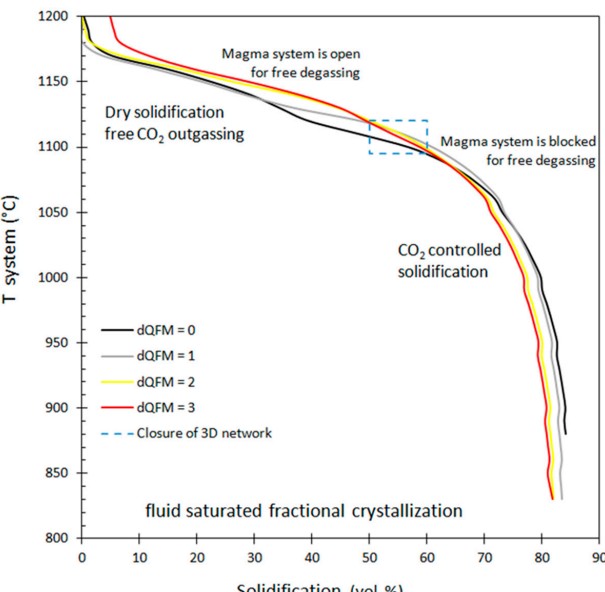

**Figure 14.** Solidification development of a basalt with Les-1 composition (after correction for $CaCO_3$) based on temperature and for different redox $\Delta$QFM conditions. The dashed field box between 1120 and 1090 °C corresponds to a critical limit of 50–60 vol.% for dyke propagation and magma eruption [113,114]. After reaching 50 vol.% solidification, the degassed fluid was retained in a closed network of interconnected crystals, which blocked continuous venting of the system and slowed the conductive cooling of the crystallizing magma. The ternary feldspars composition corresponds to up to 50% fractionation, after which the Pl composition changed toward Ab.

### 3.5. Zircon Geochronology, Lu-Hf Contents, and Ti-in-Zircon Thermometry

Separated zircon grains from the Les-1 dolerite dyke are euhedral to subhedral and are mostly polyzonal, with oscillatory zonation, typical of magmatic zircons. The grain size is from ~80 to 250 µm, but mostly around 100 µm (Figure 15). A total of 14 SIMS U-Pb spot analyses were performed on the Les-1 sample (Figure 16; Table 2). The $f_{206}$ values (proportion of common $^{206}$Pb to total measured $^{206}$Pb) of all data ranged from 0.07 to 21.76%. The zircon crystals contain 335–774 ppm U and 110–523 ppm Th, with a Th/U ratio ranging from 0.31 to 0.68 (mean 0.44). The U-Pb concordia age of the magmatic (autocrystic) population is 12.69 ± 0.24 Ma (MSWD of concordance = 0.14) (Figure 16). The remainder of the analyses yielded older ages of 389.4 ± 5.7 Ma (2σ) to 647.2 ± 17.8 Ma (2σ) that are clearly inherited. This inherited population contains 443–2822 ppm U and 61–367 ppm Th, with a Th/U ratio ranging from 0.07 to 0.26 (mean 0.14).

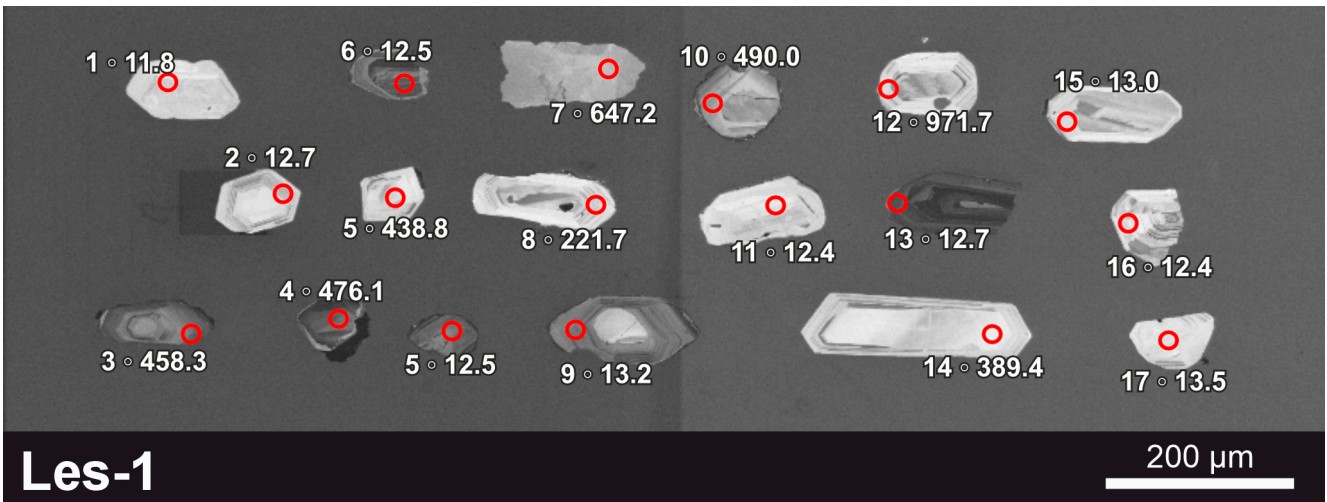

**Figure 15.** Zircon CL images from the SRM dolerite dykes with dated spots by SIMS (sample Les-1).

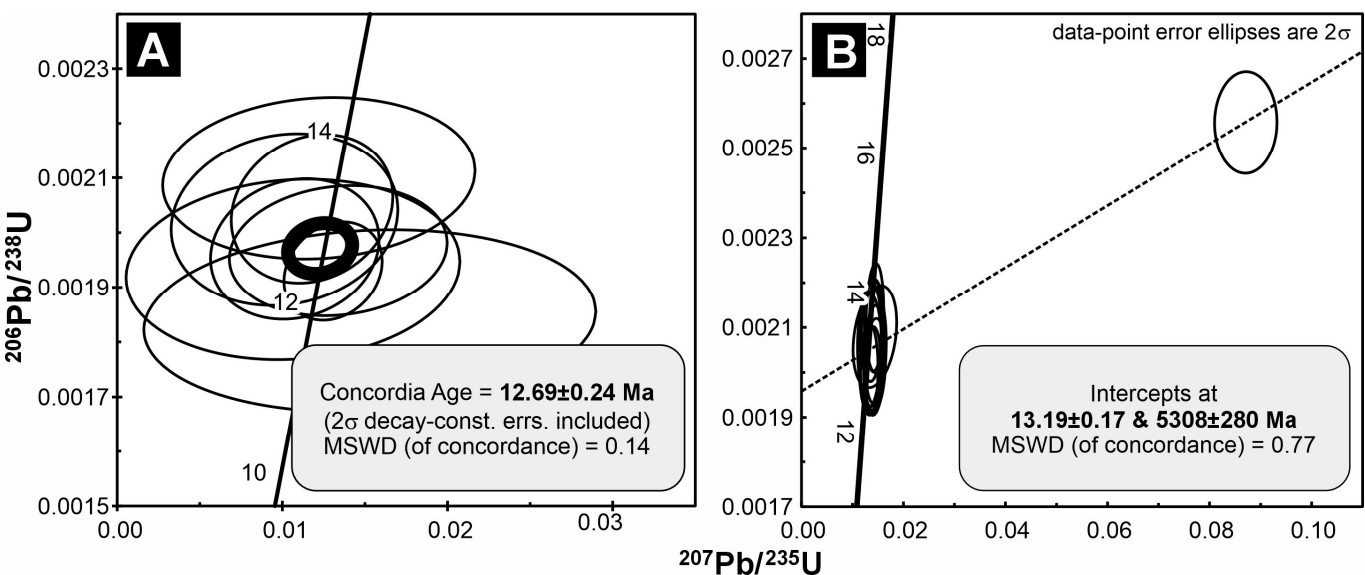

**Figure 16.** Zircon U/Pb concordias from the dolerite dykes in the Lesnica Valley. (**A**) U/Pb SIMS age diagram for sample Les-1. (**B**) U/Pb LA-ICP-MS age diagram for sample Les-2.

**Table 2.** Zircon U/Pb SIMS age data for the Les-1 dolerite dyke sample.

| Spot | U (ppm) | Th/U | $f_{206}$ (%) | $^{207}Pb/^{235}U$ | ±σ (%) | $^{206}Pb/^{238}U$ | ±σ (%) | ρ | $^{207}Pb/^{235}U$ Age (Ma) | ±σ (Ma) | $^{206}Pb/^{238}U$ Age (Ma) | ±σ (Ma) |
|---|---|---|---|---|---|---|---|---|---|---|---|---|
| 01 | 402 | 0.366 | 21.76 | 0.01529 | 36.55 | 0.0018 | 3.69 | 0.10100 | 15.4 | 5.6 | 11.8 | 0.4 |
| 02 | 695 | 0.364 | 5.76 | 0.01075 | 19.41 | 0.0020 | 2.66 | 0.13713 | 10.9 | 2.1 | 12.7 | 0.3 |
| 03 | 719 | 0.088 | 0.36 | 0.60511 | 1.93 | 0.0782 | 1.50 | 0.78011 | 480.5 | 7.4 | 485.3 | 7.0 |
| 04 | 1546 | 0.068 | 0.11 | 0.60184 | 1.63 | 0.0767 | 1.50 | 0.91969 | 478.4 | 6.2 | 476.1 | 6.9 |
| 05 | 740 | 0.083 | 0.52 | 0.53949 | 2.08 | 0.0704 | 1.50 | 0.72051 | 438.1 | 7.4 | 438.8 | 6.4 |
| 07 | 1430 | 0.256 | 0.18 | 0.89458 | 3.04 | 0.1056 | 2.89 | 0.95120 | 648.9 | 14.7 | 647.2 | 17.8 |
| 09 | 631 | 0.451 | 3.76 | 0.01179 | 16.95 | 0.0020 | 2.71 | 0.15993 | 11.9 | 2.0 | 13.2 | 0.4 |
| 10 | 443 | 0.222 | 0.20 | 0.63436 | 1.98 | 0.0790 | 1.51 | 0.76037 | 498.8 | 7.8 | 490.0 | 7.1 |
| 11 | 335 | 0.329 | 7.55 | 0.01064 | 38.79 | 0.0019 | 3.48 | 0.08976 | 10.8 | 4.2 | 12.4 | 0.4 |
| 13 | 652 | 0.571 | 9.12 | 0.01338 | 20.13 | 0.0020 | 2.47 | 0.12261 | 13.5 | 2.7 | 12.7 | 0.3 |
| 14 | 2822 | 0.124 | 0.07 | 0.47236 | 1.58 | 0.0623 | 1.50 | 0.94965 | 392.8 | 5.2 | 389.4 | 5.7 |
| 15 | 384 | 0.305 | 6.30 | 0.01013 | 27.68 | 0.0020 | 3.17 | 0.11447 | 10.2 | 2.8 | 13.0 | 0.4 |
| 16 | 774 | 0.676 | 1.60 | 0.01307 | 9.30 | 0.0019 | 1.90 | 0.20474 | 13.2 | 1.2 | 12.4 | 0.2 |
| 17 | 533 | 0.473 | 8.55 | 0.01221 | 31.65 | 0.0021 | 2.88 | 0.09084 | 12.3 | 3.9 | 13.5 | 0.4 |

A total of 15 spot LA-ICP-MS U-Pb zircon analyses were performed on the Les-2 sample (Figure 16; Table 3). The $f_{206}$ values of all data ranged from 0.17 to 25.91%. The zircon crystals contain 218–622 ppm U and 71–357 ppm Th, with a Th/U ratio ranging from 0.33 to 0.57 (mean 0.47). The U-Pb concordia age of the magmatic population is 13.19 ± 0.17 Ma (MSWD of concordance = 0.77). Three analyses yielded older ages of 84.8 ± 4.9 Ma (2σ) to 643.1 ± 30.6 Ma (2σ) interpreted as inherited zircon, with 60–311 ppm U, 21–147 ppm Th, and a Th/U ratio ranging from 0.40 to 0.47 (mean 0.42).

**Table 3.** Zircon U/Pb LA-ICP-MS age data for the Les-2 dolerite dyke sample.

| Spot | U (ppm) | Th/U | $f_{206}$ | $^{207}Pb/^{235}U$ | ±σ | $^{206}Pb/^{238}U$ | ±σ | ρ | $^{207}Pb/^{235}U$ Age (Ma) | ±σ (Ma) | $^{206}Pb/^{238}U$ Age (Ma) | ±σ (Ma) |
|---|---|---|---|---|---|---|---|---|---|---|---|---|
| 01 | 590 | 0.561 | 0.003426 | 0.0135 | 0.0016 | 0.002010 | 0.000075 | 0.2640 | 13.6 | 1.6 | 12.9 | 0.5 |
| 02 | 486 | 0.430 | 0.003553 | 0.0133 | 0.0014 | 0.002007 | 0.000083 | 0.0112 | 13.4 | 1.4 | 12.9 | 0.5 |
| 03 | 59.5 | 0.398 | 0.002099 | 0.7770 | 0.0390 | 0.092700 | 0.003000 | 0.6234 | 583.8 | 29.3 | 571.5 | 18.5 |
| 04 | 52.9 | 0.399 | 0.001692 | 0.8840 | 0.0420 | 0.103900 | 0.003400 | 0.1680 | 643.1 | 30.6 | 637.2 | 20.9 |
| 05 | 435 | 0.390 | 0.007095 | 0.0143 | 0.0017 | 0.002027 | 0.000100 | 0.1528 | 14.4 | 1.7 | 13.1 | 0.6 |
| 06 | 542 | 0.504 | 0.005184 | 0.0143 | 0.0016 | 0.002104 | 0.000085 | 0.0010 | 14.4 | 1.6 | 13.5 | 0.5 |
| 07 | 622 | 0.574 | 0.006840 | 0.0143 | 0.0015 | 0.002040 | 0.000089 | 0.1048 | 14.4 | 1.5 | 13.1 | 0.6 |
| 08 | 429 | 0.499 | 0.004927 | 0.0144 | 0.0014 | 0.002123 | 0.000098 | 0.1293 | 14.5 | 1.4 | 13.7 | 0.6 |
| 09 | 311 | 0.473 | 0.259101 | 0.0871 | 0.0050 | 0.002557 | 0.000093 | 0.0118 | 84.8 | 4.9 | 16.5 | 0.6 |
| 10 | 383 | 0.499 | 0.005187 | 0.0144 | 0.0035 | 0.002080 | 0.000093 | 0.2723 | 14.5 | 3.5 | 13.4 | 0.6 |
| 11 | 218 | 0.325 | 0.005824 | 0.0138 | 0.0023 | 0.002060 | 0.000110 | 0.0390 | 13.9 | 2.3 | 13.3 | 0.7 |
| 12 | 267 | 0.341 | 0.005447 | 0.0135 | 0.0019 | 0.002040 | 0.000110 | 0.0010 | 13.6 | 1.9 | 13.1 | 0.7 |
| 13 | 416 | 0.548 | 0.002908 | 0.0137 | 0.0016 | 0.002087 | 0.000088 | 0.0836 | 13.8 | 1.6 | 13.4 | 0.6 |
| 14 | 434 | 0.535 | 0.008236 | 0.0144 | 0.0017 | 0.002017 | 0.000084 | 0.1395 | 14.5 | 1.7 | 13.0 | 0.5 |
| 15 | 543 | 0.457 | 0.003160 | 0.0138 | 0.0018 | 0.002094 | 0.000093 | 0.1269 | 13.9 | 1.8 | 13.5 | 0.6 |

The Les-2 magmatic zircon population contains between 4.0 and 6.1 ppm of Ti (Table 4). Resultant Ti-in-Zrn crystallization temperatures after [115] are 666–699 °C (mean 676 °C). Inherited zircons show higher values of Ti between 7.5 and 18.3 ppm with temperatures between 716 and 797 °C (mean 754 °C). Lu and Hf values of the magmatic zircons are 49–68 ppm and 8750–10,820 ppm, respectively (ratio 0.0050–0.0068). Inherited zircon crystals have 45–80 ppm Lu and 9740–10,460 ppm Hf (ratio 0.0043–0.0077) (Table 4).

**Table 4.** Zircon Lu, Hf, and Ti elemental concentrations (LA-ICP-MS) data for the Les-2 dolerite dyke and Ti-in-zircon temperatures afterward [115].

| Spot | Ti (ppm) | ±σ | Lu (ppm) | ±σ | Hf (ppm) | ±σ | Lu/Hf | T (°C) |
|---|---|---|---|---|---|---|---|---|
| 01 | 6.1 | 1.6 | 49.4 | 2.5 | 8750 | 440 | 0.0056 | 699 |
| 02 | 4.2 | 1.2 | 51.8 | 2.4 | 9430 | 400 | 0.0055 | 670 |
| 03 | 7.5 | 1.5 | 74.9 | 4.9 | 9740 | 480 | 0.0077 | 716 |

**Table 4.** *Cont.*

| Spot | Ti | $\pm\sigma$ | Lu | $\pm\sigma$ | Hf | $\pm\sigma$ | Lu/Hf | T |
|---|---|---|---|---|---|---|---|---|
| | (ppm) | | (ppm) | | (ppm) | | | °C |
| 04 | 10.9 | 2 | 80.4 | 3.3 | 10,450 | 510 | 0.0077 | 748 |
| 05 | 4.8 | 1.3 | 56 | 3.5 | 9820 | 430 | 0.0057 | 680 |
| 06 | 5.1 | 1.5 | 68.1 | 4.9 | 10,080 | 520 | 0.0068 | 685 |
| 07 | 4.3 | 1.4 | 60.3 | 4.9 | 10,150 | 630 | 0.0059 | 672 |
| 08 | 4.5 | 1.4 | 53.8 | 4 | 10,450 | 490 | 0.0051 | 675 |
| 09 | 18.3 | 4.3 | 44.5 | 2.9 | 10,460 | 500 | 0.0043 | 797 |
| 10 | 5.7 | 2.6 | 52.7 | 5.8 | 9900 | 1100 | 0.0053 | 694 |
| 11 | 2.8 | 1.1 | 53.9 | 3.6 | 10,820 | 500 | 0.0050 | 640 |
| 12 | 5.8 | 1.4 | 65.1 | 3.5 | 10,470 | 510 | 0.0062 | 695 |
| 13 | 4 | 1.2 | 60.3 | 3 | 10,150 | 510 | 0.0059 | 666 |
| 14 | 4.1 | 1.3 | 62.6 | 3.5 | 10,350 | 530 | 0.0060 | 668 |
| 15 | 4.3 | 1.2 | 62.6 | 2.9 | 10,370 | 470 | 0.0060 | 672 |

## 4. Discussion

### 4.1. Parental Melt Conditions and Evolution

The commonly observed subophitic, intersertal, and porphyric textures show a transition into a micropoikilitic texture with either the same assemblage (Cpx-Ab-Mag) or that of Cal-Chl-Act±Ttn, Ep, Mag. The Cal-Chl-Act-Ttn±Ep, Mag assemblage in basalts implies subsolidus alteration. Their mutual contacts are sharp or diffusive, with Cal and Chl filling in the interstitial space. The relationship between the micropoikilitic Cpx-Ab-Mag and the spherically isometric Cal-Chl indicates liquid–liquid ("L-L") or liquid–gas ("L-G") contacts (Figure 3E–G). Therefore, the Cal and Chl in interstitial textures, amygdales, or "oceli" formed during the ongoing solidification stage in the presence of a hydrated residual silicate melt. The morphological transition from isometric to columnar and skeletal habits of the Cpx, Pl, and Fe-Ti oxides were directly controlled by the degree of magma supercooling due to heat conduction into the surrounding cold sedimentary rocks and their degassing history. The small volume of subvolcanic rocks (magma fingers of propagating feeder dykes) relative to the host sedimentary rocks maximizes the undercooling effect, which is reflected in the crystal habits of the Cpx, Pl, and Fe-Ti oxides.

Although $H_2O$ and $CO_2$ exert a fundamental control on phase relationships in magmas, at near-surface conditions, it is negligible due to the low solubility of $H_2O$ and $CO_2$ [116]. In shallow intrusions, it is assumed $H_2O+CO_2$ represents bubbles in a three-phase system L+S+G. The Cal amygdales are evidence of the presence of a free fluid phase degassing at low pressure. However, a blocked degassing fluid in the crystal network affects solidification because the thermal conductivity of gases is generally much lower than that of a liquid or solid phase. The blocked bubbles thus inhibit conductive cooling of the intrusion and prolong solidification.

The increase in oxidation state after magmatic emplacement in a complex of sedimentary rocks is expected. The increase in $Fe^{3+}/Fe^{2+}$ is recorded by Cpx, Fe-Ti oxides TiMag → Mag, Hem, or by Ep crystallization in the subsolidus assemblage Chl-Act-Ttn-Mag-Cal-Ep. Oxidation is related to the degassing of trapped $CO_2$ in the solidifying dolerite crystal network, such as the reaction $2Fe^{2+} + CO_2 + H_2O \rightarrow CO + 2Fe^{3+} + 2OH^-$ proposed by [75].

The presence of albitic Pl in subvolcanic and (particularly) volcanic basaltic rocks is generally considered to be a product of syn-volcanic alteration by residual fluids or post-magmatic interaction with hydrothermal fluids [77]. The onset of skeletal Ab crystallization as a primary magmatic mineral is expected at T = 1120 °C by rhyolite-MELTS simulations, which corresponds to the composition of intermediate Pl relics. At 1120 °C, the simulations show that when crystals comprise more than 50% by volume of the magma–crystal mush, solidification starts blocking the degassed $CO_2$ bubbles escaping from the magma. All Ab is without incongruent precipitates of clinozoisite (Czo), Prh, böhmite (Bhm), or Qz.

Albite with an epitaxial relationship with Cpx (Figure 3B) or Ab of skeletal habit both imply a magmatic origin. The absence (both theoretically and petrographically) of secondary phases associated with Ab is strongly against the formation of Ab as a decomposition product of more basic magmatic Pl.

The SRM rocks lie on subalkaline and alkaline trends, with a spectrum of trachy-basalts, basaltic andesites, to basaltic trachyandesites, which correspond to differentiation at 1180–1100 °C from the parental Les-1 magma. The differentiation trends are consistent with rhyolite-MELTS simulations, which respected several chemical parameters for the fractionated melt, including $SiO_2$, $Na_2O+K_2O$, and $Al_2O_3/TiO_2$ under variable redox conditions ($\Delta QFM = 0$ to $\Delta QFM = 3$) and saturated for $H_2O+CO_2$ (Table 1). The chemical differences between the samples constrain solidification to 1130–1100 °C, i.e., the temperature range at which it reaches 50%–60% crystals by volume, the point at which magma loses its ability to flow through magma channels or a feeder dyke system [111–113]. Simulations of fractional and equilibrium crystallization under low-pressure conditions ($p = 10$ bar) for $\Delta QFM = 0$ to $\Delta QFM = +3$ showed only limited $H_2O$ (0.44–0.18 wt.%) and $CO_2$ ($5.2 \times 10^{-3}$–$0.2 \times 10^{-3}$ wt.%) in the residual melt. The ability of a solidifying system to maintain the residual fluid was controlled by the surrounding wall rocks as well as the crystal network.

Rhyolite-MELTS simulations suggest that observed differentiation trends (Figure 9) can only be explained by magma fractionation of the parental Les-1 dolerite with the removal of $CaCO_3$. Carbonate assimilation on the fractionation path is supported by the composition of rock-forming minerals such as Cpx, Pl, and Ab. Carbonate assimilation from the wall rocks likely resulted in the formation of alkaline varieties of trachybasalt and basaltic trachyandesite from a subalkaline tholeiite basaltic magma. Ti-in-zircon thermometry shows a relatively low temperature for zircon crystallization (666–699 °C), which may have resulted from the presence of carbonatitic melt driving the system to relatively lower subsolidus temperatures (see Section 4.2).

*4.2. Presence of Carbonatitic Melt*

The zonal fabric within some intersertal spaces shows very efficient fractionation of the micropoikilitic Ab-Cpx-Mag assemblage from Chl+Cal. Micropoikilitic aggregates of the undercooled silicate melt do not contain Chl or Cal, and their boundaries are sharp (Figure 3E–G). Similarly, the Chl+Cal spheres do not contain minerals present in the micropoikilitic aggregates. This situation may indicate the existence of separate silicate Ab-Cpx-Mag and carbonatitic Chl?-Cal melts, which at some point became immiscible.

The origin of the carbonate melt preserved in the amygdaloids or "oceli" space is related to the fractionation of intruded magma accompanied by the assimilation of carbonate material incorporated and dispersed into the host magma during the eruption. The basaltic rock chemistry indicates simultaneous fractionation of liquid subalkaline and alkaline lines of descent (Figure 9) of a parental magma composition (Les-1 with $CaCO_3$ removal) with $\Delta QFM = 0$ to +2. The presence of $CaCO_3$ prior to intrusion during magma fractionation would lead to fractionation lines of descent, which do not match the composition of the samples in Figure 9. Similarly, the Ab composition of feldspars cannot be explained by progressive crystallization because the feldspars of intermediate composition (50%–10% An) are missing. The only way to explain the suppression of the crystallization of An-Pl is the isolation of calcium from the differentiating silicate melt into the immiscible carbonatitic melt. The coexistence of both types of melts and their spatially isolated solidification products of Cpx-Ab-Mag and Cal-Chl are identified in the zoned structures of "oceli" or amygdaloids (Figure 3E–G). Carbonate-bearing amygdales have been reported to crystallize from immiscible carbonatitic–basaltic melts in [117–119].

Effective separation of the low-viscous carbonatitic melt from the silicate interstitial melt into the central portion of the spherical microbodies (Figure 4) might be related to the process of dynamic extraction of intergranular melts, which is controlled by the movement of escaping bubbles from the blocking crystal network. Without a free fluid phase, such extraction dynamics would not be possible. The existence of separate batches of melt

requires specific conditions under which the differentiating melt reaches immiscibility between silicate and carbonatite melt. Petrological experiments focused on reconstructing the coexistence of silicate and carbonate melt are extensive [120–122], but these are not easily applied to low-pressure systems, such as the SRM volcanic–intrusive complex, which additionally represents an open magmatic system (e.g., controlled by external supercooling, pulsative degassing, or contemporaneous assimilation).

### 4.3. Effect of Limestone Assimilation on the Basaltic Rock Composition

The influence of the assimilation of surrounding limestones on the chemical composition of basaltic rocks is evident from field, petrographic, EPMA microchemical, and geochemical studies. These include fragments of limestone identified in lava (Figure 2A) together with zircon xenocrysts, positive $\Delta Zr$ anomalies, excess Ca linked to the presence of pure $CaCO_3$, positive HFSE anomalies, geochemical coupling of $CaCO_3$ and $^{87}Sr/^{86}Sr$, and high $^{206}Pb/^{204}Pb$ associated with high Pb concentrations. However, the uniform and depleted Nd isotopic compositions ($\varepsilon Nd_{(13)}$ values of +8.0 to +7.4) together with low Th (b.d.l.) and LREE depletion indicate that assimilated limestone had very low REE, Th, and LILE contents in comparison to typical crustal material (e.g., clastic sediments and granitic rocks), which are characterized by high $La/Yb_n$, $Th/La_n$, and LILE contents and typically negative $\varepsilon Nd$ values [123–125].

The fragments of incorporated carbonate (Figure 2A) might be the product of physical shattering by magma penetration into the shallow subsurface along with decompression degassing, e.g., [126–128]. The effect of $CaCO_3$ assimilation on the fractionation path of basalt is recorded by Cpx, with an increase in Di-Hd together with an increase in $Fe^{3+}$ and $Na^+$, as shown by experimental studies [70–73], although some inferred pre-assimilation Cpx does not show the effects of $CaCO_3$ assimilation (Figure 5).

### 4.4. Geodynamic Analogues of the Lesnica Volcanic-Intrusive Complex

The basaltic rocks in the SRM are aligned along the Lesnica Valley, close to the Muráň fault. The search (using the GEOROCK [129] and EARTHCHEM [130] databases) for geochemically analogous basalts revealed similar basalts ($La/Sm_n < 1$, $Sm/Yb_n > 1$, and $La/Yb_n < 1$) from the transform faults environments, such as the Blanco transform fault (e.g., [96]; Figure 11). Other geodynamic analogs to the basaltic rocks from the Lesnica Valley include Carboniferous basalts from the Agishan-Yamansu belt in NW China [131], which have similar REE systematics, namely $La/Yb_n < 1$ (0.41–1.11), $La/Sm_n < 1$ (0.36–0.96), and $Sm/Yb_n > 1$ (1.13–1.37). The Eastern Tianshan Agishan-Yamansu volcanos are spatially bound to significant fault zones, which may further indicate a systematic link between basalts with $La/Sm_n < 1$ and $Sm/Yb_n > 1$ and transtensional fault structures.

The N-MORB-like REE and trace element distributions (Figures 11 and 12) imply large degrees of partial melting. However, the regional geological setting precludes large-scale extension or seafloor spreading being involved in the genesis of the SRM basalts. The alignment of the feeder dykes along the Lesnica Valley may be related to the strike-slip, NE–SW trending Muráň fault (Figure 1A). This NE–SW orientation is the same as the NE–SW maximum compression orientation followed by extension inferred in the Middle/Late Miocene [10,11,49], which overlaps with the timing of emplacement of the basaltic dykes from our geochronological data. Moreover, our geochronological data support the migration of volcanic activity from west to east (Section 1.2).

Magmatism in the Lesnica Valley was synchronous with the formation of the southern intracontinental basins in the early Middle Miocene, where orogen-parallel extension in the IWC resulted in the hinterland (back arc) basin system formation between ca. 15 and 13 Ma [10,11,13]. At his time, movement took place on the Muráň transtensional fault, accompanying obliquely oriented structures (Figure 1B), which facilitated the emplacement of the basaltic magmas in the Lesnica Valley.

### 5. Conclusions

1.  A basaltic dyke swarm was intruded into sandstones and carbonates of the Triassic Bódvaszilas Formation of the Silica Nappe in Lesnica Valley (Slovenský Raj Mountains, IWC);

2.  We provide evidence for late magmatic carbonatitic (Cal-bearing) "oceli" formation. They grew within the silicate Cpx-Pl-TiMag network and were overgrown by a micropoikilitic Cpx-Ab-TiMag assemblage located between the "oceli" and the outer Cpx-Pl-TiMag ring-shaped zone. Subsolidus Chl-Act-Cal-Ep-Ttn aggregates partly to totally overgrew the magmatic assemblages.

3.  The N-MORB-like volcanic rocks are tholeiitic-alkaline ($La/Yb_n$ and $La/Sm_n < 1$, $Sm/Yb_n$ and $Gd/Yb_n \geq 1$ at near constant $Eu/Eu^*$ of ~0.9) with $\varepsilon Nd_{(13)}$ = +8.0 to +7.4;

4.  Rhyolite-MELTS simulations show that the erupted products are the result of magmatic differentiation of a parental basaltic tholeiitic magma with $\Delta QFM$ = +1 to +3, affected by varying degrees (0%–50%) of fractionation and assimilation of carbonate material in a shallow magmatic reservoir. The REE values were modeled by 1% fractional melting of garnet peridotite mixed in a 1:9 ratio with 7% melting of spinel peridotite of PM composition;

5.  SIMS and LA-ICP-MS U-Pb zircon dating of two dolerite dykes yields Miocene (Serravallian) ages of $12.69 \pm 0.24$ Ma and $13.3 \pm 0.16$ Ma. Magmatism in the Slovenský Raj Mountains corresponds to the end of the main phase of calc-alkaline andesite volcanism in the central Slovakia volcanic field and its shift towards calc-alkaline volcanism in the Slanské vrchy and Vihorlat mountains in eastern Slovakia;

6.  The 13.3–12.7 Ma magmatic activity in the Lesnica Valley was synchronous with the early Middle Miocene orogen-parallel extension in the IWC and hinterland (back arc) basin system formation. The isolated basaltic magmatism shows that it was limited, was probably pulsed, and was not associated with significant intracontinental spreading. Basalt feeder dyke emplacement in the Lesnica Valley was facilitated by the Muráň transtensional fault, but the ultimate sourcing of asthenospheric melts was likely linked to deep-seated structures with shallow magmatic reservoirs along the major Cenozoic ALCAPA (Alps–Carpathians–Pannonia) microplate and EP (European plate) boundary.

**Supplementary Materials:** The following supporting information can be downloaded at https://www.mdpi.com/article/10.3390/min14010009/s1, Table S1: Analytical data.

**Author Contributions:** Conceptualization, R.D. and M.P.; methodology, R.D., Q.-L.L., L.A. and D.C.; software, R.D., Q.-L.L., L.A. and D.C.; validation, R.D. and M.P.; formal analysis, R.D., M.P., O.N., Q.-L.L., L.A. and D.C.; investigation, R.D., Q.-L.L., L.A. and D.C.; resources, R.D., Q.-L.L., L.A. and D.C.; data curation, R.D., Q.-L.L., L.A. and D.C.; writing—original draft preparation, R.D.; writing—review and editing, R.D., M.P., Q.-L.L., L.A., D.C. and O.N.; visualization, R.D. and O.N.; supervision, M.P., Q.-L.L., L.A. and D.C.; project administration, M.P., Q.-L.L., L.A. and D.C.; funding acquisition, M.P., Q.-L.L., L.A. and D.C. All authors have read and agreed to the published version of the manuscript.

**Funding:** This research was funded by The Slovak Research and Development Agency APVV-19-0065 (M.P.), the Chinese (2016YFE0203000, Q.-L.L.), and the Czech (RVO67985831, L.A.) scientific grants. D. Chew acknowledges support from Science Foundation Ireland through research grants 12/IP/1663, 13/RC/2092, and 13/RC/2092_P2 (iCRAG Research Centre).

**Data Availability Statement:** https://www.geology.sk/sluzby/digitalny-archiv/ (accessed on 24 October 2023).

**Acknowledgments:** R.D. greatly acknowledges the National Park of Slovenský Raj for the permission to carry out the field investigation. We thankfully acknowledge the comments and suggestions of the reviewers, which significantly improved the original manuscript.

**Conflicts of Interest:** The authors declare no conflict of interest.

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
