# Peer review of "Miocene Volcanism in the Slovenský Raj Mountains: Magmatic, Space, and Time Relationships in the Western Carpathians"

_minerals, doi:10.3390/min14010009_

Round 1

Reviewer 1 Report

Comments and Suggestions for Authors

Subject of this study is a few basalt outcrops of middle Slovakia. It appears that these basalts are from outcrops with very poor exposures that make a clear volcanological assessment difficult as these were thought before this investigation just to represent “boulders” in limestones. The authors assess these basalts to represent basalt dikes that cut limestones and sandstones. In fact, there are only 5 of these 1-m sized exposures that are the topic of this paper! Basalt rocks have a lot of secondary minerals and authors acknowledge that bulk compositions have gained CaO and correct for that. I have annotated the pdf in many places but stopped later on because I would have needed a longer review time to do a thorough job of the whole manuscript. Below is my brief assessment in bullet points for brevity.

·      The manuscript has lots of data and big ideas which is great.

·      Problem is, given the nature of the basalts (minor exposures, altered composition), I worry that authors overinterpret their data in numerous places.

·      It appears to me that basalts also gained Na2O, not only CaO. Major element composition is in conflict with depleted incompatible trace element composition

·      The Results have too many interpretations already and all those should be moved to the Discussion

·      The manuscript has too many abbreviations that are nowhere explained.

·      The writing should be simpler. The sentences are often too complex and make it difficult to follow

·      Sometimes you use big words like “dynamic solidification” which does not give more infos than simpler terms; use simpler language.

·      I doubt that any of the zircons you dated crystallized from these basalts. I’m not aware of any basalt compositionally similar to your basalts to have crystallized such large and complex zircons.

·      Bulk data should be normalized to 100% for the table and pre-normalization totals given, only normalized data should be plotted

Comments on the Quality of English Language

Extensive editing of English language required

Author Response

pdf submitted

Reviewer 2 Report

Comments and Suggestions for Authors

Review of "Miocene volcanism in the Slovenský Raj Mountains: Magmatic, space and time relationships in the Western Carpathians” by Rastislav Demko et al.

The article presents new major element, geochemical, isotopic, mineral and age data on magmatic rocks of the volcanic-intrusive complex in the Slovenský Raj Mountains. The dataset appears to be of good quality and deserves to be published in the journal “Minerals”. However, the data interpretation appears to be weak (but it is rather a matter of personal taste) and the article is poorly organized. Thus, it should be accepted only after major revision. My comments to the authors are below.

Major comments:

1)      The introduction section consists basically of 5 lines, which is not enough. You should use the introduction to explain why it is important for the scientific community to study this object.

2)      The “results” section contains a lot of interpretation. Interpretation should be separated and moved to discussion. For example, I do not agree with the majority of the interpretation, but I acknowledge the new data, so it would be nice to have a concise and straightforward “Results” section.

3)      It seems like the strange behavior of LILEs is due to their strong mobility in fluids during assimilation and post-magmatic alteration. I think that you shouldn’t assume “variable external input of LILE” it rather internal re-distribution and partial loss. Such features are common for altered rocks (e.g., Popov et al., 2019).

4)      A bit of modeling would be good for isotope diagrams. For example, can you estimate the composition of the limestones and model their mixing with the most primitive of your rocks? It seems that you need to numerically explain the Sr isotopic shift with the absence of one for Nd and compare the results with other major elements and geochemical features of the rocks.

5)      The article is too long. It does not have that much new information, so it could have at least 2 times less word count.

Minor comments:

26 (and in other parts of the text) – eNd(13) – do you mean that it is eNd 13 million years ago? If this is true, you should rather use “eNdi” (initial) to not confuse the reader.

Materials and Methods. You should start this part with the number of collected samples, their short description (why they were chosen for the study) and a reference to the supplementary material. Also, please provide the sampling coordinates for them in the supplementary.

Section 3.1 will be better placed at the start of materials and methods.

Results. Please keep the discussion and results separate. Provide only facts in results and use them as argument in discussion.

Lines 313-316, 330-344, 385-397, 429-438, 445-448, 458-472, 595-603, 674-703 – seems more like parts of the discussion.

604-626 – discussion and results completely mixed.

Section 3.4. – you should describe the rock compositions (including geochemistry and isotopes) first. Typically, the rock composition should go before minerals in the “results” section. Also, this whole section combined with the discussion about geochemistry belongs to the “Discussion”

Section 3.5.2. It’s too big and misplaced. You need to significantly cut it and it definitely should be in the discussion. In my opinion, this chapter adds nearly nothing interesting to the article (you just speculating about the results of modeling with a lot of assumptions), so maybe it will be better to place it in a supplementary section and use only the most essential conclusions in the discussion. Furthermore, the modeling in some places contradicts your own petrographic data so you make more assumptions, which is not a good sign.

References:

Popov, D. V., Brovchenko, V. D., Nekrylov, N. A., Plechov, P. Y., Spikings, R. A., Tyutyunnik, O. A., ... & Soloviev, A. V. (2019). Removing a mask of alteration: Geochemistry and age of the Karadag volcanic sequence in SE Crimea. Lithos, 324, 371-384.

Comments on the Quality of English Language

The quality of the English is acceptable, but the text needs to be spellchecked. 

Author Response

pdf submitted

Round 2

Reviewer 1 Report

Comments and Suggestions for Authors

I would have had only stylistic comments such as there are so many abbreviations that is makes is hard for the reader.  I still do not necessarily agree with some of conclusions, but if the authors feel strongly about them I'm fine with them.

Reviewer 2 Report

Comments and Suggestions for Authors

The response to the review seems formal and insufficient, however, who am I to judge?

I recommend accepting the manuscript in its present form because it is unlikely that another round of review will help improve it.